# Practical Equivariances via Relational Conditional Neural Processes

**Daolang Huang**[1] **Manuel Haussmann**[1] **Ulpu Remes**[2] **ST John**[1]

**Grégoire Clarté**[3] **Kevin Sebastian Luck**[4,6] **Samuel Kaski**[1,5] **Luigi Acerbi**[3]

[1]Department of Computer Science, Aalto University, Finland
[2]Department of Mathematics and Statistics, University of Helsinki
[3]Department of Computer Science, University of Helsinki
[4]Department of Electrical Engineering and Automation (EEA), Aalto University, Finland
[5]Department of Computer Science, University of Manchester
[6]Department of Computer Science, Vrije Universiteit Amsterdam, The Netherlands
{daolang.huang, manuel.haussmann, ti.john, samuel.kaski}@aalto.fi
k.s.luck@vu.nl
{ulpu.remes, gregoire.clarte, luigi.acerbi}@helsinki.fi

## Abstract

Conditional Neural Processes (CNPs) are a class of metalearning models popular for combining the runtime efficiency of amortized inference with reliable uncertainty quantification. Many relevant machine learning tasks, such as in spatio-temporal modeling, Bayesian Optimization and continuous control, inherently contain equivariances – for example to translation – which the model can exploit for maximal performance. However, prior attempts to include equivariances in CNPs do not scale effectively beyond two input dimensions. In this work, we propose Relational Conditional Neural Processes (RCNPs), an effective approach to incorporate equivariances into any neural process model. Our proposed method extends the applicability and impact of equivariant neural processes to higher dimensions. We empirically demonstrate the competitive performance of RCNPs on a large array of tasks naturally containing equivariances.

## 1 Introduction

Conditional Neural Processes (CNPs; [10]) have emerged as a powerful family of metalearning models, offering the flexibility of deep learning along with well-calibrated uncertainty estimates and a tractable training objective. CNPs can naturally handle irregular and missing data, making them suitable for a wide range of applications. Various advancements, such as attentive (ACNP; [20]) and Gaussian (GNP; [30]) variants, have further broadened the applicability of CNPs. In principle, CNPs can be trained on other general-purpose stochastic processes, such as Gaussian Processes (GPs; [34]), and be used as an amortized, drop-in replacement for those, with minimal computational cost at runtime.

However, despite their numerous advantages, CNPs face substantial challenges when attempting to model equivariances, such as translation equivariance, which are essential for problems involving spatio-temporal components or for emulating widely used GP kernels in tasks such as Bayesian Optimization (BayesOpt; [12]). In the context of CNPs, kernel properties like stationarity and isotropy would correspond to, respectively, translational equivariance and equivariance to rigid transformations. Lacking such equivariances, CNPs struggle to scale effectively and emulate (equivariant) GPs even in moderate higher-dimensional input spaces (i.e., above two). Follow-up work has introduced Convolutional CNPs (ConvCNP; [15]), which leverage a convolutional deep sets construction to

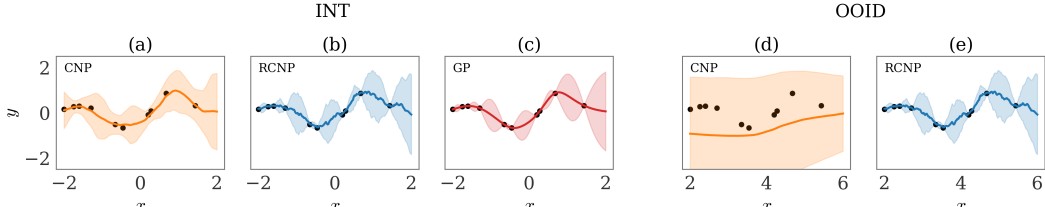

Figure 1: **Equivariance in 1D regression.** Left: Predictions for a CNP (a) and RCNP (b) in an *interpolation* (INT) task, trained for $20$ epochs to emulate a GP (c) with Matérn-$\frac{5}{2}$ kernel and noiseless observations. The CNP underfits the context data (black dots), while the RCNP leverages translation equivariance to learn faster and yield better predictions. Right: The CNP (d) fails to predict in an *out-of-input-distribution* (OOID) task, where the input context is outside the training range (note the shifted $x$ axis); whereas the RCNP (e) generalizes by means of translational equivariance.

induce translational-equivariant embeddings. However, the requirement of defining an input grid and performing convolutions severely limits the applicability of ConvCNPs and variants thereof (ConvGNP [30]; FullConvGNP [2]) to one- or two-dimensional equivariant inputs; both because higher-dimensional implementations of convolutions are poorly supported by most deep learning libraries, and for the prohibitive cost of performing convolutions in three or more dimensions. Thus, the problem of efficiently scaling equivariances in CNPs above two input dimensions remains open.

In this paper, we introduce Relational Conditional Neural Processes (RCNPs), a novel approach that offers a simple yet powerful technique for including a large class of equivariances into any neural process model. By leveraging the existing equivariances of a problem, RCNPs can achieve improved sample efficiency, predictive performance, and generalization (see Figure 1). The basic idea in RCNPs is to enforce equivariances via a relational encoding that only stores appropriately chosen *relative* information of the data. By stripping away absolute information, equivariance is automatically satisfied. Surpassing the complex approach of previous methods (e.g., the ConvCNP family for translational equivariance), RCNPs provide a practical solution that scales to higher dimensions, while maintaining strong performance and extending to other equivariances. The cost to pay is increased computational complexity in terms of context size (size of the dataset we are conditioning on at runtime); though often not a bottleneck for the typical metalearning small-context setting of CNPs. Our proposed method works for equivariances that can be expressed relationally via comparison between pairs of points (e.g., their difference or distance); in this paper, we focus on translational equivariance and equivariance to rigid transformations.

**Contributions.** In summary, our contributions in this work are:

- We introduce a simple and effective way – *relational encoding* – to encode exact equivariances directly into CNPs, in a way that easily scales to higher input dimensions.
- We propose two variants of relational encoding: one that works more generally ('Full'); and one which is simpler and more computationally efficient ('Simple'), and is best suited for implementing translation equivariance.
- We provide theoretical foundations and proofs to support our approach.
- We empirically demonstrate the competitive performance of RCNPs on a variety of tasks that naturally contain different equivariances, highlighting their practicality and effectiveness.

**Outline of the paper.** The remainder of this paper is organized as follows. In Section 2, we review the foundational work of CNPs and their variants. This is followed by the introduction of our proposed relational encoding approach to equivariances at the basis of our RCNP models (Section 3). We then provide in Section 4 theoretical proof that relational encoding achieves equivariance without losing essential information; followed in Section 5 by a thorough empirical validation of our claims in various tasks requiring equivariances, demonstrating the generalization capabilities and predictive performance of RCNPs. We discuss other related work in Section 6, and the limitations of our approach, including its computational complexity, in Section 7. We conclude in Section 8 with an overview of the current work and future directions.

Our code is available at https://github.com/acerbilab/relational-neural-processes.

## 2 Background: the Conditional Neural Process family

In this section, we review the Conditional Neural Process (CNP) family of stochastic processes and the key concept of equivariance at the basis of this work. Following [30], we present these notions within the framework of prediction maps [8]. We denote with $\mathbf{x} \in \mathcal{X} \subseteq \mathbb{R}^{d_x}$ input vectors and $\mathbf{y} \in \mathcal{Y} \subseteq \mathbb{R}^{d_y}$ output vectors, with $d_x, d_y \geq 1$ their dimensionality. If $f(\mathbf{z})$ is a function that takes as input elements of a set $\mathbf{Z}$, we denote with $f(\mathbf{Z})$ the set $\{f(\mathbf{z})\}_{\mathbf{z} \in \mathbf{Z}}$.

**Prediction maps.** A *prediction map* $\pi$ is a function that maps (1) a *context set* $(\mathbf{X}, \mathbf{Y})$ comprising input/output pairs $\{(\mathbf{x}_1, \mathbf{y}_1), \dots, (\mathbf{x}_N, \mathbf{y}_N)\}$ and (2) a collection of *target inputs* $\mathbf{X}^\star = (\mathbf{x}_1^\star, \dots, \mathbf{x}_M^\star)$ to a distribution over the corresponding *target outputs* $\mathbf{Y}^\star = (\mathbf{y}_1^\star, \dots, \mathbf{y}_M^\star)$:

$$\pi(\mathbf{Y}^\star | (\mathbf{X}, \mathbf{Y}), \mathbf{X}^\star) = p(\mathbf{Y}^\star | \mathbf{r}), \tag{1}$$

where $\mathbf{r} = r((\mathbf{X}, \mathbf{Y}), \mathbf{X}^\star)$ is the *representation vector* that parameterizes the distribution over $\mathbf{Y}^\star$ via the representation function $r$. Bayesian posteriors are prediction maps, a well-known example being the Gaussian Process (GP) posterior:

$$\pi(\mathbf{Y}^\star | (\mathbf{X}, \mathbf{Y}), \mathbf{X}^\star) = \mathcal{N}\left(\mathbf{Y}^\star | \mathbf{m}, \mathbf{K}\right), \tag{2}$$

where the prediction map takes the form of a multivariate normal with representation vector $\mathbf{r} = (\mathbf{m}, \mathbf{K})$. The mean $\mathbf{m} = m_{\text{post}}((\mathbf{X}, \mathbf{Y}), \mathbf{X}^\star)$ and covariance matrix $\mathbf{K} = k_{\text{post}}(\mathbf{X}, \mathbf{X}^\star)$ of the multivariate normal are determined by the conventional GP posterior predictive expressions [34].

**Equivariance.** A prediction map $\pi$ with representation function $r$ is $\mathcal{T}$-*equivariant* with respect to a group $\mathcal{T}$ of transformations[1] of the input space, $\tau : \mathcal{X} \rightarrow \mathcal{X}$, if and only if for all $\tau \in \mathcal{T}$:

$$r((\mathbf{X}, \mathbf{Y}), \mathbf{X}^\star) = r((\tau\mathbf{X}, \mathbf{Y}), \tau\mathbf{X}^\star), \tag{3}$$

where $\tau\mathbf{x} \equiv \tau(\mathbf{x})$ and $\tau\mathbf{X}$ is the set obtained by applying $\tau$ to all elements of $\mathbf{X}$. Eq. 3 defines equivariance of a prediction map based on its representation function, and can be shown to be equivalent to the common definition of an *equivariant map*; see Appendix A. Intuitively, equivariance means that if the data (the context inputs) are transformed in a certain way, the predictions (the target inputs) transform correspondingly. Common groups of transformations include translations, rotations, reflections – all examples of rigid transformations. In kernel methods and specifically in GPs, equivariances are incorporated in the prior kernel function $k(\mathbf{x}, \mathbf{x}^\star)$. For example, translational equivariance corresponds to *stationarity* $k_{\text{sta}} = k(\mathbf{x} - \mathbf{x}^\star)$, and equivariance to all rigid transformations corresponds to *isotropy*, $k_{\text{iso}} = k(||\mathbf{x} - \mathbf{x}^\star||_2)$, where $|| \cdot ||_2$ denotes the Euclidean norm of a vector. A crucial question we address in this work is how to implement equivariances in other prediction maps, and specifically in the CNP family.

**Conditional Neural Processes.** A CNP [10] uses an *encoder*[2] $f_e$ to produce an embedding of the context set, $\mathbf{e} = f_e(\mathbf{X}, \mathbf{Y})$. The encoder uses a DeepSet architecture [48] to ensure invariance with respect to permutation of the order of data points, a key property of stochastic processes. We denote with $\mathbf{r}_m = (\mathbf{e}, \mathbf{x}_m^\star)$ the local representation of the $m$-th point of the target set $\mathbf{X}^\star$, for $1 \leq m \leq M$. CNPs yield a prediction map with representation $\mathbf{r} = (\mathbf{r}_1, \dots, \mathbf{r}_M)$:

$$\pi(\mathbf{Y}^\star | (\mathbf{X}, \mathbf{Y}), \mathbf{X}^\star) = p(\mathbf{Y}^\star | \mathbf{r}) = \prod_{m=1}^{M} q\left(\mathbf{y}_m^\star | \boldsymbol{\lambda}(\mathbf{r}_m)\right), \tag{4}$$

where $q(\cdot | \boldsymbol{\lambda})$ belongs to a family of distributions parameterized by $\boldsymbol{\lambda}$, and $\boldsymbol{\lambda} = f_d(\mathbf{r}_m)$ is decoded in parallel for each $\mathbf{r}_m$. In the standard CNP, the decoder network $f_d$ is a multi-layer perceptron. A common choice for CNPs is a Gaussian likelihood, $q(\mathbf{y}_m^\star | \boldsymbol{\lambda}) = \mathcal{N}\left(\mathbf{y}_m^\star | \boldsymbol{\mu}(\mathbf{r}_m), \boldsymbol{\Sigma}(\mathbf{r}_m)\right)$, where $\boldsymbol{\mu}$ and $\boldsymbol{\Sigma}$ represent the predictive mean and covariance of each output, independently for each target (a *mean field* approach). Given the closed-form likelihood, CNPs are easily trainable via maximum-likelihood optimization of parameters of encoder and decoder networks, by sampling batches of context and target sets from the training data.

---

[1] A group of transformations is a family of composable, invertible functions with identity $\tau_{\text{id}} \mathbf{x} = \mathbf{x}$.

[2] We use purple to highlight parametrized functions (neural networks) whose parameters will be learned.

**Gaussian Neural Processes.** Notably, standard CNPs do not model dependencies between distinct target outputs $\mathbf{y}_m^\star$ and $\mathbf{y}_{m'}^\star$, for $m \neq m'$. Gaussian Neural Processes (GNPs [30]) are a variant of CNPs that remedy this limitation, by assuming a joint multivariate normal structure over the outputs for the target set, $\pi(\mathbf{Y}^\star | (\mathbf{X}, \mathbf{Y}), \mathbf{X}^\star) = \mathcal{N}(\mathbf{Y} | \boldsymbol{\mu}, \boldsymbol{\Sigma})$. For ease of presentation, we consider now scalar outputs ($d_y = 1$), but the model generalizes to the multi-output case. GNPs parameterize the mean as $\mu_m = f_\mu(\mathbf{r}_m)$ and covariance matrix $\Sigma_{m,m'} = k\big(f_\Sigma(\mathbf{r}_m), f_\Sigma(\mathbf{r}_{m'})\big) f_v(\mathbf{r}_m) f_v(\mathbf{r}_{m'})$, for target points $\mathbf{x}_m^\star, \mathbf{x}_{m'}^\star$, where $f_\mu$, $f_\Sigma$, and $f_v$ are neural networks with outputs, respectively, in $\mathbb{R}$, $\mathbb{R}^{d_\Sigma}$, and $\mathbb{R}^+$, $k(\cdot, \cdot)$ is a positive-definite kernel function, and $d_\Sigma \in \mathbb{N}^+$ denotes the dimensionality of the space in which the covariance kernel is evaluated. Standard GNP models use the `linear` covariance (where $f_v = 1$ and $k$ is the linear kernel) or the `kvv` covariance (where $k$ is the exponentiated quadratic kernel with unit lengthscale), as described in [30].

**Convolutional Conditional Neural Processes.** The Convolutional CNP family includes the ConvCNP [15], ConvGNP [30], and FullConvGNP [2]. These CNP models are built to implement translational equivariance via a ConvDeepSet architecture [15]. For example, the ConvCNP is a prediction map $p(\mathbf{Y}^\star | (\mathbf{X}, \mathbf{Y}), \mathbf{X}^\star) = \prod_{m=1}^{M} q(\mathbf{y}_m^\star | \Phi_{\mathbf{X}, \mathbf{Y}}(\mathbf{x}_m^\star))$, where $\Phi_{\mathbf{X}, \mathbf{Y}}(\cdot)$ is a ConvDeepSet. The construction of ConvDeepSets involves, among other steps, gridding of the data if not already on the grid and application of $d_x$-dimensional convolutional neural networks ($2d_x$ for FullConv-GNP). Due to the limited scaling and availability of convolutional operators above two dimensions, ConvCNPs do not scale in practice for $d_x > 2$ translationally-equivariant input dimensions.

**Other Neural Processes.** The neural process family includes several other members, such as latent NPs (LNP; [11]) which model dependencies in the predictions via a latent variable – however, LNPs lack a tractable training objective, which impairs their practical performance. Attentive (C)NPs (A(C)NPs; [20]) implement an attention mechanism instead of the simpler DeepSet architecture. Transformer NPs [31] combine a transformer-based architecture with a causal mask to construct an autoregressive likelihood. Finally, Autoregressive CNPs (AR-CNPs [3]) provide a novel technique to deploy existing CNP models via autoregressive sampling without architectural changes.

## 3    Relational Conditional Neural Processes

We introduce now our Relational Conditional Neural Processes (RCNPs), an effective solution for embedding equivariances into any CNP model. Through relational encoding, we encode selected relative information and discard absolute information, inducing the desired equivariance.

**Relational encoding.** In RCNPs, the (full) *relational encoding* of a target point $\mathbf{x}_m^\star \in \mathbf{X}^\star$ with respect to the context set $(\mathbf{X}, \mathbf{Y})$ is defined as:

$$\rho_{\text{full}}(\mathbf{x}_m^\star, (\mathbf{X}, \mathbf{Y})) = \bigoplus_{n,n'=1}^{N} f_r\left(g(\mathbf{x}_n, \mathbf{x}_m^\star), \mathbf{R}_{nn'}\right), \qquad \mathbf{R}_{nn'} \equiv (g(\mathbf{x}_n, \mathbf{x}_{n'}), \mathbf{y}_n, \mathbf{y}_{n'}), \quad (5)$$

where $g : \mathcal{X} \times \mathcal{X} \rightarrow \mathbb{R}^{d_{\text{comp}}}$ is a chosen *comparison function*[3] that specifies how a pair $\mathbf{x}, \mathbf{x}'$ should be compared; $\mathbf{R}$ is the *relational matrix*, comparing all pairs of the context set; $f_r : \mathbb{R}^{d_{\text{comp}}} \times \mathbb{R}^{d_{\text{comp}}+2d_y} \rightarrow \mathbb{R}^{d_{\text{rel}}}$ is the *relational encoder*, a neural network that maps a comparison vector and element of the relational matrix into a high-dimensional space $\mathbb{R}^{d_{\text{rel}}}$; $\bigoplus$ is a commutative aggregation operation (`sum` in this work) ensuring permutation invariance of the context set [48]. From Eq. 5, a point $\mathbf{x}^\star$ is encoded based on how it compares to the entire context set.

Intuitively, the comparison function $g(\cdot, \cdot)$ should be chosen to remove all information that does not matter to impose the desired equivariance. For example, if we want to encode translational equivariance, the comparison function should be the difference of the inputs, $g_{\text{diff}}(\mathbf{x}_n, \mathbf{x}_m^\star) = \mathbf{x}_m^\star - \mathbf{x}_n$ (with $d_{\text{comp}} = d_x$). Similarly, isotropy (invariance to rigid transformations, i.e. rotations, translations, and reflections) can be encoded via the Euclidean distance $g_{\text{dist}}(\mathbf{x}_n, \mathbf{x}_m^\star) = ||\mathbf{x}_m^\star - \mathbf{x}_n||_2$ (with $d_{\text{comp}} = 1$). We will prove these statements formally in Section 4.

---

[3]We use green to highlight the selected comparison function that encodes a specific set of equivariances.

**Full RCNP.** The full-context RCNP, or FullRCNP, is a prediction map with representation $\mathbf{r} = (\boldsymbol{\rho}_1, \ldots, \boldsymbol{\rho}_M)$, with $\boldsymbol{\rho}_m = \rho_{\text{full}}(\mathbf{x}_m^\star, (\mathbf{X}, \mathbf{Y}))$ the relational encoding defined in Eq. 5:

$$\pi(\mathbf{Y}^\star | (\mathbf{X}, \mathbf{Y}), \mathbf{X}^\star) = p(\mathbf{Y}^\star | \mathbf{r}) = \prod_{m=1}^{M} q\left(\mathbf{y}_m^\star | \boldsymbol{\lambda}(\boldsymbol{\rho}_m)\right), \tag{6}$$

where $q(\cdot | \boldsymbol{\lambda})$ belongs to a family of distributions parameterized by $\boldsymbol{\lambda}$, where $\boldsymbol{\lambda} = f_d(\boldsymbol{\rho}_m)$ is decoded from the relational encoding $\boldsymbol{\rho}_m$ of the $m$-th target. As usual, we often choose a Gaussian likelihood, whose mean and covariance (variance, for scalar outputs) are produced by the decoder network.

Note how Eq. 6 (FullRCNP) is nearly identical to Eq. 4 (CNP), the difference being that we replaced the representation $\mathbf{r}_m = (\mathbf{e}, \mathbf{x}_m^\star)$ with the relational encoding $\boldsymbol{\rho}_m$ from Eq. 5. Unlike CNPs, in RCNPs there is no separate encoding of the context set alone. The RCNP construction generalizes easily to other members of the CNP family by plug-in replacement of $\mathbf{r}_m$ with $\boldsymbol{\rho}_m$. For example, a relational GNP (RGNP) describes a multivariate normal prediction map whose mean is parameterized as $\mu_m = f_\mu(\boldsymbol{\rho}_m)$ and whose covariance matrix is given by $\Sigma_{m,m'} = k\left(f_\Sigma(\boldsymbol{\rho}_m), f_\Sigma(\boldsymbol{\rho}_{m'})\right) f_v(\boldsymbol{\rho}_m) f_v(\boldsymbol{\rho}_{m'})$.

**Simple RCNP.** The full relational encoding in Eq. 5 is cumbersome as it asks to build and aggregate over a full relational matrix. Instead, we can consider the simple or 'diagonal' relational encoding:

$$\rho_{\text{diag}}\left(\mathbf{x}_m^\star, (\mathbf{X}, \mathbf{Y})\right) = \bigoplus_{n=1}^{N} f_r\left(g(\mathbf{x}_n, \mathbf{x}_m^\star), g(\mathbf{x}_n, \mathbf{x}_n), \mathbf{y}_n\right). \tag{7}$$

Eq. 7 is functionally equivalent to Eq. 5 restricted to the diagonal $n = n'$, and further simplifies in the common case $g(\mathbf{x}_n, \mathbf{x}_n) = \mathbf{0}$, whereby the argument of the aggregation becomes $f_r(g(\mathbf{x}_n, \mathbf{x}_m^\star), \mathbf{y}_n)$.

We obtain the simple RCNP model (from now on, just RCNP) by using the diagonal relational encoding $\rho_{\text{diag}}$ instead of the full one, $\rho_{\text{full}}$. Otherwise, the simple RCNP model follows Eq. 6. We will prove, both in theory and empirically, that the simple RCNP is best for encoding translational equivariance. Like the FullRCNP, the RCNP easily extends to other members of the CNP family.

In this paper, we consider the FullRCNP, FullRGNP, RCNP and RGNP models for translations and rigid transformations, leaving examination of other RCNP variants and equivariances to future work.

## 4 RCNPs are equivariant and context-preserving prediction maps

In this section, we demonstrate that RCNPs are $\mathcal{T}$-equivariant prediction maps, where $\mathcal{T}$ is a transformation group of interest (e.g., translations), for an appropriately chosen comparison function $g : \mathcal{X} \times \mathcal{X} \to \mathbb{R}^{d_{\text{comp}}}$. Then, we formalize the statement that RCNPs strip away only enough information to achieve equivariance, but no more. We prove this by showing that the RCNP representation preserves information in the context set. Full proofs are given in Appendix A.

### 4.1 RCNPs are equivariant

**Definition 4.1.** Let $g$ be a comparison function and $\mathcal{T}$ a group of transformations $\tau : \mathcal{X} \to \mathcal{X}$. We say that $g$ is $\mathcal{T}$-*invariant* if and only if $g(\mathbf{x}, \mathbf{x}') = g(\tau\mathbf{x}, \tau\mathbf{x}')$ for any $\mathbf{x}, \mathbf{x}' \in \mathcal{X}$ and $\tau \in \mathcal{T}$.

**Definition 4.2.** Given a comparison function $g$, we define the *comparison sets*:

$$g\left((\mathbf{X}, \mathbf{Y}), (\mathbf{X}, \mathbf{Y})\right) = \left\{(g(\mathbf{x}_n, \mathbf{x}_{n'}), \mathbf{y}_n, \mathbf{y}_{n'})\right\}_{1 \leq n, n' \leq N},$$
$$g\left((\mathbf{X}, \mathbf{Y}), \mathbf{X}^\star\right) = \left\{(g(\mathbf{x}_n, \mathbf{x}_m^\star), \mathbf{y}_n)\right\}_{1 \leq n \leq N, 1 \leq m \leq M}, \tag{8}$$
$$g\left(\mathbf{X}^\star, \mathbf{X}^\star\right) = \left\{g(\mathbf{x}_m^\star, \mathbf{x}_{m'}^\star)\right\}_{1 \leq m, m' \leq M}.$$

If $g$ is not symmetric, we can also denote $g\left(\mathbf{X}^\star, (\mathbf{X}, \mathbf{Y})\right) = \left\{(g(\mathbf{x}_m^\star, \mathbf{x}_n), \mathbf{y}_n)\right\}_{1 \leq n \leq N, 1 \leq m \leq M}$.

**Definition 4.3.** A prediction map $\pi$ and its representation function $r$ are *relational* with respect to a comparison function $g$ if and only if $r$ can be written solely through set comparisons:

$$r((\mathbf{X}, \mathbf{Y}), \mathbf{X}^\star) = r\left(g((\mathbf{X}, \mathbf{Y}), (\mathbf{X}, \mathbf{Y})), g((\mathbf{X}, \mathbf{Y}), \mathbf{X}^\star), g(\mathbf{X}^\star, (\mathbf{X}, \mathbf{Y})), g(\mathbf{X}^\star, \mathbf{X}^\star)\right). \tag{9}$$

**Lemma 4.4.** *Let $\pi$ be a prediction map, $\mathcal{T}$ a transformation group, and $g$ a comparison function. If $\pi$ is relational with respect to $g$ and $g$ is $\mathcal{T}$-invariant, then $\pi$ is $\mathcal{T}$-equivariant.*

From Lemma 4.4 and previous definitions, we derive the main result about equivariance of RCNPs.

**Proposition 4.5.** *Let $g$ be the comparison function used in a RCNP, and $\mathcal{T}$ a group of transformations. If $g$ is $\mathcal{T}$-invariant, the RCNP is $\mathcal{T}$-equivariant.*

As useful examples, the *difference* comparison function $g_{\text{diff}}(\mathbf{x}, \mathbf{x}^{\star}) = \mathbf{x}^{\star} - \mathbf{x}$ is invariant to translations of the inputs, and the *distance* comparison function $g_{\text{dist}}(\mathbf{x}, \mathbf{x}^{\star}) = ||\mathbf{x}^{\star} - \mathbf{x}||_2$ is invariant to rigid transformations; thus yielding appropriately equivariant RCNPs.

### 4.2 RCNPs are context-preserving

The previous section demonstrates that any RCNP is $\mathcal{T}$-equivariant, for an appropriate choice of $g$. However, a trivial comparison function $g(\mathbf{x}, \mathbf{x}') = \mathbf{0}$ would also satisfy the requirements, yielding a trivial representation. We need to guarantee that, at least in principle, the encoding procedure removes only information required to induce $\mathcal{T}$-equivariance, and no more. A minimal request is that the context set is preserved in the prediction map representation $\mathbf{r}$, modulo equivariances.

**Definition 4.6.** A comparison function $g$ is *context-preserving* with respect to a transformation group $\mathcal{T}$ if for any context set $(\mathbf{X}, \mathbf{Y})$ and target set $\mathbf{X}^{\star}$, there is a submatrix $\mathbf{Q}' \subseteq \mathbf{Q}$ of the matrix $\mathbf{Q}_{mnn'} = (g(\mathbf{x}_n, \mathbf{x}_m^{\star}), g(\mathbf{x}_n, \mathbf{x}_{n'}), \mathbf{y}_n, \mathbf{y}_{n'})$, a reconstruction function $\gamma$, and a transformation $\tau \in \mathcal{T}$ such that $\gamma(\mathbf{Q}') = (\tau \mathbf{X}, \mathbf{Y})$.

For example, $g_{\text{dist}}$ is context-preserving with respect to the group of rigid transformations. For any $m$, $\mathbf{Q}' = \mathbf{Q}_{m::}$ is the set of pairwise distances between points, indexed by their output values. Reconstructing the positions of a set of points given their pairwise distances is known as the *Euclidean distance geometry* problem [38], which can be solved uniquely up to rigid transformations with traditional multidimensional scaling techniques [39]. Similarly, $g_{\text{diff}}$ is context-preserving with respect to translations. For any $m$ and $n'$, $\mathbf{Q}' = \mathbf{Q}_{m:n'}$ can be projected to the vector $((\mathbf{x}_1 - \mathbf{x}_m^{\star}, \mathbf{y}_1), \ldots, (\mathbf{x}_N - \mathbf{x}_m^{\star}, \mathbf{y}_N))$, which is equal to $(\tau_m \mathbf{X}, \mathbf{Y})$ with the translation $\tau_m(\cdot) = \cdot - \mathbf{x}_m^{\star}$.

**Definition 4.7.** For any $(\mathbf{X}, \mathbf{Y}), \mathbf{X}^{\star}$, a family of functions $h_{\boldsymbol{\theta}}((\mathbf{X}, \mathbf{Y}), \mathbf{X}^{\star}) \to \mathbf{r} \in \mathbb{R}^{d_{\text{rep}}}$ is *context-preserving* under a transformation group $\mathcal{T}$ if there exists $\boldsymbol{\theta} \in \boldsymbol{\Theta}$, $d_{\text{rep}} \in \mathbb{N}$, a *reconstruction function* $\gamma$, and a transformation $\tau \in \mathcal{T}$ such that $\gamma(h_{\boldsymbol{\theta}}((\mathbf{X}, \mathbf{Y}), \mathbf{X}^{\star})) \equiv \gamma(\mathbf{r}) = (\tau \mathbf{X}, \mathbf{Y})$.

Thus, an encoding is context-preserving if it is possible at least in principle to fully recover the context set from $\mathbf{r}$, implying that no relevant context is lost. This is indeed the case for RCNPs.

**Proposition 4.8.** *Let $\mathcal{T}$ be a transformation group and $g$ the comparison function used in a FullRCNP. If $g$ is context-preserving with respect to $\mathcal{T}$, then the representation function $r$ of the FullRCNP is context-preserving with respect to $\mathcal{T}$.*

**Proposition 4.9.** *Let $\mathcal{T}$ be the translation group and $g_{\text{diff}}$ the difference comparison function. The representation of the simple RCNP model with $g_{\text{diff}}$ is context-preserving with respect to $\mathcal{T}$.*

Given the convenience of the RCNP compared to FullRCNPs, Proposition 4.9 shows that we can use simple RCNPs to incorporate translation-equivariance with no loss of information. However, the simple RCNP model is *not* context-preserving for other equivariances, for which we ought to use the FullRCNP. Our theoretical results are confirmed by the empirical validation in the next section.

## 5 Experiments

In this section, we evaluate the proposed relational models on several tasks and compare their performance with other conditional neural process models. For this, we used publicly available reference implementations of the `neuralprocesses` software package [1, 3]. We detail our experimental approach in Appendix B, and we empirically analyze computational costs in Appendix C.

### 5.1 Synthetic Gaussian and non-Gaussian functions

We first provide a thorough comparison of our methods with other CNP models using a diverse array of Gaussian and non-Gaussian synthetic regression tasks. We consider tasks characterized by functions derived from (i) a range of GPs, where each GP is sampled using one of three different kernels (Exponentiated Quadratic (EQ), Matérn-$\frac{5}{2}$, and Weakly-Periodic); (ii) a non-Gaussian sawtooth

process; (iii) a non-Gaussian mixture task. In the mixture task, the function is randomly selected from either one of the three aforementioned distinct GPs or the sawtooth process, each chosen with probability $\frac{1}{4}$. Apart from evaluating simple cases with $d_x = \{1, 2\}$, we also expand our experiments to higher dimensions, $d_x = \{3, 5, 10\}$. In these higher-dimensional scenarios, applying ConvCNP and ConvGNP models is not considered feasible. We assess the performance of the models in two distinct ways. The first one, *interpolation* (INT), uses the data generated from a range identical to that employed during the training phase. The second one, *out-of-input-distribution* (OOID), uses data generated from a range that extends beyond the scope of the training data.

**Results.** We first compare our translation-equivariant ('stationary') versions of RCNP and RGNP with other baseline models from the CNP family (Table 1). Comprehensive results, including all five regression problems and five dimensions, are available in Appendix D. Firstly, relational encoding of the translational equivariance intrinsic to the task improves performance, as both RCNP and RGNP models surpass their CNP and GNP counterparts in terms of INT results. Furthermore, the OOID results demonstrate significant improvement of our models, as they can leverage translational-equivariance to generalize outside the training range. RCNPs and RGNPs are competitive with convolutional models (ConvCNP, ConvGNP) when applied to 1D data and continue performing well in higher dimension, whereas models in the ConvCNP family are inapplicable for $d_x > 2$.

Table 1: Comparison of *interpolation* (INT) and *out-of-input-distribution* (OOID) performance of our RCNP models and CNP baselines on synthetic regression tasks with varying input dimensions. We show mean and (standard deviation) across 10 runs with different seeds. "F" denotes failed attempts that yielded very bad results. Missing entries could not be run. Statistically significantly best results are **bolded**. Our methods (RCNP and RGNP) perform better than their CNP and GNP counterparts in terms of both INT and OOID, and scale to higher dimension compared to ConvCNP and ConvGNP.

| | | Weakly-periodic KL divergence($\downarrow$) | | | Sawtooth log-likelihood($\uparrow$) | | | Mixture log-likelihood($\uparrow$) | | |
|---|---|---|---|---|---|---|---|---|---|---|
| | | $d_x = 1$ | $d_x = 3$ | $d_x = 5$ | $d_x = 1$ | $d_x = 3$ | $d_x = 5$ | $d_x = 1$ | $d_x = 3$ | $d_x = 5$ |
| INT | RCNP (sta) | 0.24 (0.00) | 0.28 (0.00) | 0.31 (0.00) | 3.03 (0.06) | 0.85 (0.01) | 0.44 (0.00) | 0.20 (0.01) | -0.10 (0.00) | -0.31 (0.03) |
| | RGNP (sta) | 0.03 (0.00) | **0.05** (0.00) | **0.08** (0.00) | 3.90 (0.09) | **1.09** (0.01) | **1.13** (0.05) | 0.34 (0.03) | **0.37** (0.01) | **0.04** (0.02) |
| | ConvCNP | 0.21 (0.00) | - | - | 3.64 (0.04) | - | - | 0.38 (0.02) | - | - |
| | ConvGNP | **0.01** (0.00) | - | - | **3.94** (0.11) | - | - | **0.49** (0.15) | - | - |
| | CNP | 0.31 (0.00) | 0.39 (0.00) | 0.42 (0.00) | 2.25 (0.02) | 0.36 (0.28) | -0.03 (0.10) | 0.01 (0.01) | -0.57 (0.11) | -0.72 (0.08) |
| | GNP | 0.06 (0.00) | 0.08 (0.01) | 0.11 (0.01) | 0.83 (0.04) | 0.23 (0.13) | 0.02 (0.05) | 0.17 (0.01) | -0.17 (0.00) | -0.32 (0.00) |
| OOID | RCNP (sta) | 0.24 (0.00) | 0.28 (0.01) | 0.31 (0.00) | 3.04 (0.06) | 0.85 (0.01) | 0.44 (0.00) | 0.20 (0.01) | -0.10 (0.00) | -0.31 (0.03) |
| | RGNP (sta) | 0.03 (0.00) | **0.05** (0.01) | **0.08** (0.00) | 3.90 (0.10) | **1.09** (0.01) | **1.13** (0.05) | 0.34 (0.03) | **0.37** (0.01) | **0.04** (0.02) |
| | ConvCNP | 0.21 (0.00) | - | - | 3.64 (0.04) | - | - | 0.38 (0.02) | - | - |
| | ConvGNP | **0.01** (0.00) | - | - | **3.97** (0.08) | - | - | **0.49** (0.15) | - | - |
| | CNP | 2.88 (0.91) | 1.58 (0.50) | 2.20 (0.81) | F | -0.37 (0.12) | -0.22 (0.03) | F | -2.55 (1.15) | -1.71 (0.55) |
| | GNP | F | 1.47 (0.27) | 0.62 (0.04) | F | F | F | F | -0.67 (0.05) | -0.72 (0.03) |

We further consider two GP tasks with isotropic EQ and Matérn-$\frac{5}{2}$ kernels (invariant to rigid transformations). Within this set of experiments, we include the FullRCNP and FullRGNP models, each equipped with the 'isotropic' distance comparison function. The results (Table 2) indicate that RCNPs and FullRCNPs consistently outperform CNPs across both tasks. Additionally, we notice that FullRCNPs exhibit better performance compared to RCNPs as the dimension increases. When $d_x = 2$, the performance of our RGNPs is on par with that of ConvGNPs, and achieves the best results in terms of both INT and OOID when $d_x > 2$, which again highlights the effectiveness of our models in handling high-dimensional tasks by leveraging existing equivariances.

## 5.2 Bayesian optimization

We explore the extent our proposed models can be used for a higher-dimensional meta-learning task, using Bayesian optimization (BayesOpt) as our application [12]. The neural processes, ours as well as the baselines, serve as surrogates to find the global minimum $f_{\min} = f(\mathbf{x}_{\min})$ of a black-box function. For this task, we train the models by generating random functions from a GP kernel sampled from a set of base kernels—EQ, Matérn-$\{\frac{1}{2}, \frac{3}{2}, \frac{5}{2}\}$ as well as their sums and products—with randomly sampled hyperparameters. By training on a large distribution over kernels, we aim to exploit the metalearning capabilities of neural processes. The trained CNP models are then used as surrogates to minimize the Hartmann function [40, p.185] in three and six dimensions, a common BayesOpt test function. We use the *expected improvement* acquisition function, which we can evaluate analytically. Specifics on the experimental setup and further evaluations can be found in Appendix E.

Table 2: Comparison of the *interpolation* (INT) and *out-of-input-distribution* (OOID) performance of our RCNP models with different CNP baselines on two GP synthetic regression tasks with isotropic kernels of varying input dimensions.

| | | EQ KL divergence($\downarrow$) | | | Matérn-$\frac{5}{2}$ KL divergence($\downarrow$) | | |
|---|---|---|---|---|---|---|---|
| | | $d_x = 2$ | $d_x = 3$ | $d_x = 5$ | $d_x = 2$ | $d_x = 3$ | $d_x = 5$ |
| INT | RCNP (sta) | 0.26 (0.00) | 0.40 (0.01) | 0.45 (0.00) | 0.30 (0.00) | 0.39 (0.00) | 0.35 (0.00) |
| | RGNP (sta) | 0.03 (0.00) | **0.05** (0.00) | **0.11** (0.00) | 0.03 (0.00) | **0.05** (0.00) | **0.11** (0.00) |
| | FullRCNP (iso) | 0.26 (0.00) | 0.31 (0.00) | 0.35 (0.00) | 0.30 (0.00) | 0.32 (0.00) | 0.29 (0.00) |
| | FullRGNP (iso) | 0.08 (0.00) | 0.14 (0.00) | 0.25 (0.00) | 0.09 (0.00) | 0.16 (0.00) | 0.21 (0.00) |
| | ConvCNP | 0.22 (0.00) | - | - | 0.26 (0.00) | - | - |
| | ConvGNP | **0.01** (0.00) | - | - | **0.01** (0.00) | - | - |
| | CNP | 0.33 (0.00) | 0.44 (0.00) | 0.57 (0.00) | 0.39 (0.00) | 0.46 (0.00) | 0.47 (0.00) |
| | GNP | 0.05 (0.00) | 0.09 (0.01) | 0.19 (0.00) | 0.07 (0.00) | 0.11 (0.00) | 0.19 (0.00) |
| OOID | RCNP (sta) | 0.26 (0.00) | 0.40 (0.01) | 0.45 (0.00) | 0.30 (0.00) | 0.39 (0.00) | 0.35 (0.00) |
| | RGNP (sta) | 0.03 (0.00) | **0.05** (0.00) | **0.11** (0.00) | 0.03 (0.00) | **0.05** (0.00) | **0.11** (0.00) |
| | FullRCNP (iso) | 0.26 (0.00) | 0.31 (0.00) | 0.35 (0.00) | 0.30 (0.00) | 0.32 (0.00) | 0.29 (0.00) |
| | FullRGNP (iso) | 0.08 (0.00) | 0.14 (0.00) | 0.25 (0.00) | 0.09 (0.00) | 0.16 (0.00) | 0.21 (0.00) |
| | ConvCNP | 0.22 (0.00) | - | - | 0.26 (0.00) | - | - |
| | ConvGNP | **0.01** (0.00) | - | - | **0.01** (0.00) | - | - |
| | CNP | 4.54 (1.76) | 3.30 (1.55) | 1.22 (0.09) | 6.75 (2.72) | 1.75 (0.42) | 0.93 (0.02) |
| | GNP | 2.25 (0.61) | 2.54 (1.44) | 0.74 (0.02) | 1.86 (0.26) | 1.23 (0.17) | 0.62 (0.02) |

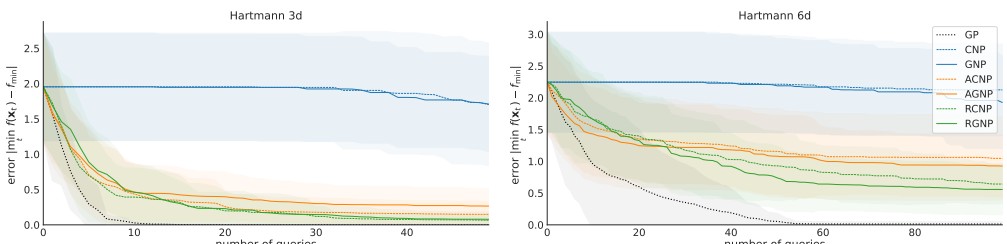

Figure 2: **Bayesian Optimization.** Error during optimization of a 3D/6D Hartmann function (lower is better). RCNP/RGNP improve upon the baselines, approaching the GP performance.

**Results.** As shown in Figure 2, RCNPs and RGNPs are able to learn from the random functions and come close to the performance of a Gaussian Process (GP), the most common surrogate model for BayesOpt. Note that the GP is refit after every new observation, while the CNP models can condition on the new observation added to the context set, without any retraining. CNPs and GNPs struggle with the diversity provided by the random kernel function samples and fail at the task. In order to remain competitive, they need to be extended with an attention mechanism (ACNP, AGNP).

## 5.3 Lotka–Volterra model

Neural process models excel in the so-called sim-to-real task where a model is trained using simulated data and then applied to real-world contexts. Previous studies have demonstrated this capability by training neural process models with simulated data generated from the stochastic Lotka–Volterra predator–prey equations and evaluating them with the famous hare–lynx dataset [32]. We run this benchmark evaluation using the simulator and experimental setup proposed by [3]. Here the CNP models are trained with simulated data and evaluated with both simulated and real data; the learning tasks represented in the training and evaluation data include *interpolation*, *forecasting*, and *reconstruction*. The evaluation results presented in Table 3 indicate that the best model depends on the task type, but overall our proposed relational CNP models with translational equivariance perform comparably to their convolutional and attentive counterparts, showing that our simpler approach does not hamper performance on real data. Full results with baseline CNPs are provided in Appendix F.

## 5.4 Reaction-Diffusion model

The Reaction-Diffusion (RD) model is a large class of state-space models originating from chemistry [45] with several applications in medicine [13] and biology [22]. We consider here a reduced model representing the evolution of cancerous cells [13], which interact with healthy cells through the

Table 3: Normalized log-likelihood scores in the Lotka–Volterra experiments (higher is better). The mean and (standard deviation) reported for each model are calculated based on 10 training outcomes evaluated with the same simulated (S) and real (R) learning tasks. The tasks include *interpolation* (INT), *forecasting* (FOR), and *reconstruction* (REC). Statistically significantly (see Appendix B.1) best results are **bolded**. RCNP and RGNP models perform on par with convolutional and attentive baselines.

|  | INT (S) | FOR (S) | REC (S) | INT (R) | FOR (R) | REC (R) |
|---|---|---|---|---|---|---|
| RCNP | -3.57 (0.02) | -4.85 (0.00) | -4.20 (0.01) | -4.24 (0.02) | -4.83 (0.03) | -4.55 (0.05) |
| RGNP | -3.51 (0.01) | **-4.27** (0.00) | -3.76 (0.00) | -4.31 (0.06) | **-4.47** (0.03) | -4.39 (0.11) |
| ConvCNP | -3.47 (0.01) | -4.85 (0.00) | -4.06 (0.00) | **-4.21** (0.04) | -5.01 (0.02) | -4.75 (0.05) |
| ConvGNP | **-3.46** (0.00) | -4.30 (0.00) | **-3.67** (0.01) | **-4.19** (0.02) | -4.61 (0.03) | -4.62 (0.11) |
| ACNP | -4.04 (0.06) | -4.87 (0.01) | -4.36 (0.03) | **-4.18** (0.05) | -4.79 (0.03) | -4.48 (0.02) |
| AGNP | -4.12 (0.17) | -4.35 (0.09) | -4.05 (0.19) | -4.33 (0.15) | **-4.48** (0.06) | **-4.29** (0.10) |

production of acid. These three quantities (healthy cells, cancerous cells, acid concentration) are defined on a discretized space-time grid (2+1 dimensions, $d_x = 3$). We assume we only observe the difference in number between healthy and cancerous cells, which makes this model a hidden Markov model. Using realistic parameters inspired by [13], we simulated $4 \cdot 10^3$ full trajectories, from which we subsample observations to generate training data for the models; another set of $10^3$ trajectories is used for testing. More details can be found in Appendix G.

We propose two tasks: first, a *completion* task, where target points at time $t$ are inferred through the context at different spatial locations at time $t-1$, $t$ and $t+1$; secondly, a *forecasting* task, where the target at time $t$ is inferred from the context at $t-1$ and $t-2$. These tasks, along with the form of the equation describing the model, induce translation invariance in both space and time, which requires the models to incorporate translational equivariance for all three dimensions.

**Results.** We compare our translational-equivariant RCNP models to ACNP, CNP, and their GNP variants in Table 4. Comparison with ConvCNP is not feasible, as this problem is three-dimensional. Our methods outperform the others on both the *completion* and *forecasting* tasks, showing the advantage of leveraging translational equivariance in this complex spatio-temporal modeling problem.

Table 4: Normalized log-likelihood scores in the Reaction-Diffusion problem for both tasks (higher is better). Mean and (standard deviation) from 10 training runs evaluated on a separate test dataset.

|  | RCNP | RGNP | ACNP | AGNP | CNP | GNP |
|---|---|---|---|---|---|---|
| Completion | 0.22 (0.33) | **1.38** (0.62 ) | 0.17 (0.03) | 0.20 (0.03) | 0.10 (0.01) | 0.13 (0.03) |
| Forecasting | **0.07** (0.18) | **-0.18** (0.58) | -0.65 (0.31) | -1.58 (1.05) | -0.51 (0.20) | -0.50 (0.30) |

### 5.5 Additional experiments

As further empirical tests of our method, we study our technique in the context of autoregressive CNPs in Appendix H.1, present a proof-of-concept of incorporating rotational symmetry in Appendix H.2, and examine the performance of RCNPs on image regression in Appendix H.3.

## 6 Related work

This work builds upon the foundation laid by CNPs [10] and other members of the CNP family, covered at length in Section 2. A significant body of work has focused on incorporating equivariances into neural network models [33, 4, 23, 41]. The concept of equivariance has been explored in Convolutional Neural Networks (CNNs; [24]) with translational equivariance, and more generally in Group Equivariant CNNs [5], where rotations and reflections are also considered. Work on DeepSets laid out the conditions for permutation invariance and equivariance [48]. Set Transformers [25] extend

this approach with an attention mechanism to learn higher-order interaction terms among instances of a set. Our work focuses on incorporating equivariances into prediction maps, and specifically CNPs.

Prior work on incorporating equivariances into CNPs requires a regular discrete lattice of the input space for their convolutional operations [15, 30]. EquivCNPs [19] build on work by [7] which operates on irregular point clouds, but they still require a constructed lattice over the input space. SteerCNPs [18] generalize ConvCNPs to other equivariances, but still suffer from the same scaling issues. These methods are therefore in practice limited to low-dimensional (one to two equivariant dimensions), whereas our proposal does not suffer from this constraint.

Our approach is also related to metric-based meta-learning, such as Prototypical Networks [36] and Relation Networks [37]. These methods learn an embedding space where classification can be performed by computing distances to prototype representations. While effective for few-shot classification tasks, they may not be suitable for more complex tasks or those requiring uncertainty quantification. GSSM [46] aims to *learn* relational biases via a graph structure on the context set, while we directly build *exact* equivariances into the CNP architecture.

Kernel methods and Gaussian processes (GPs) have long addressed issues of equivariance by customizing kernel designs to encode specific equivariances [14]. For instance, stationary kernels are used to capture globally consistent patterns [34], with applications in many areas, notably Bayesian Optimization [12]. However, despite recent computational advances [9], kernel methods and GPs still struggle with high-dimensional, complex data, with open challenges in deep kernel learning [47] and amortized kernel learning (or metalearning) [26, 35], motivating our proposal of RCNPs.

## 7 Limitations

RCNPs crucially rely on a comparison function $g(\mathbf{x}, \mathbf{x}')$ to encode equivariances. The comparison functions we described (e.g., for isotropy and translational equivariance) already represent a large class of useful equivariances. Notably, key contributions to the neural process literature focus only on translation equivariance (e.g., ConvCNP [15], ConvGNP [30], FullConvGNP [2]). Extending our method to other equivariances will require the construction of new comparison functions.

The main limitation of the RCNP class is its increased computational complexity in terms of context and target set sizes (respectively, $N$ and $M$). The FullRCNP model can be cumbersome, with $O(N^2 M)$ cost for training and deployment. However, we showed that the simple or 'diagonal' RCNP variant can fully implement translational invariance with a $O(NM)$ cost. Still, this cost is larger than $O(N + M)$ of basic CNPs. Given the typical metalearning setting of small-data regime (small context sets), the increased complexity is often acceptable, outweighed by the large performance improvement obtained by leveraging available equivariances. This is shown in our empirical validation, in which RCNPs almost always outperformed their CNP counterparts.

## 8 Conclusion

In this paper, we introduced Relational Conditional Neural Processes (RCNPs), a new member of the neural process family which incorporates equivariances through relational encoding of the context and target sets. Our method applies to equivariances that can be induced via an appropriate comparison function; here we focused on translational equivariances (induced by the difference comparison) and equivariances to rigid transformations (induced by the distance comparison). How to express other equivariances via our relational approach is an interesting direction for future work.

We demonstrated with both theoretical results and extensive empirical validation that our method successfully introduces equivariances in the CNP model class, performing comparably to the translational-equivariant ConvCNP models in low dimension, but with a simpler construction that allows RCNPs to scale to larger equivariant input dimensions ($d_x > 2$) and outperform other CNP models.

In summary, we showed that the RCNP model class provides a simple and effective way to implement translational and other equivariances into the CNP model family. Exploiting equivariances intrinsic to a problem can significantly improve performance. Open problems remain in extending the current approach to other equivariances which are not expressible via a comparison function, and making the existing relational approach more efficient and scalable to larger context datasets.

## Acknowledgments and Disclosure of Funding

This work was supported by the Academy of Finland Flagship programme: Finnish Center for Artificial Intelligence FCAI. Samuel Kaski was supported by the UKRI Turing AI World-Leading Researcher Fellowship, [EP/W002973/1]. The authors wish to thank the Finnish Computing Competence Infrastructure (FCCI), Aalto Science-IT project, and CSC–IT Center for Science, Finland, for the computational and data storage resources provided.

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

# Appendix

In this Appendix, we include our methodology, mathematical proofs, implementation details for reproducibility, additional results and extended explanations omitted from the main text.

**Contents**

## A  Theoretical proofs

In this section, we provide extended proofs for statements and theorems in the main text. First, we show that our definition of equivariance for the prediction map is equivalent to the common definition of equivalent maps (Section A.1). Then, we provide full proofs that RCNPs are equivariant (Section A.2) and context-preserving (Section A.3) prediction maps, as presented in the theorems in Section 4 of the main text.

### A.1  Definition of equivariance for prediction maps

We show that the definition of equivariance of a prediction map, which is introduced in the main text based on the invariance of the representation function (Eq. 3 in the main text), is equivalent to the common definition of equivariance for a generic mapping.

First, we need to extend the notion of group action of a group $\mathcal{T}$ into $\mathbb{R}^d$, for $d \in \mathbb{N}$. We define two additional actions, *(i)* for $\tau \in \mathcal{T}$ and $f$ a function from $\mathbb{R}^d$ into $\mathbb{R}^p$, and *(ii)* for a couple $(\mathbf{x}, \mathbf{y}) \in \mathbb{R}^d \times \mathbb{R}^p$. Following [15]:

$$\tau(\mathbf{x}, \mathbf{y}) = (\tau\mathbf{x}, \mathbf{y}), \qquad \tau f(\mathbf{x}) = f(\tau^{-1}\mathbf{x}).$$

These two definitions can be extended for $n$-uplets $\mathbf{X}$ and $\mathbf{Y}$:

$$\tau(\mathbf{X}, \mathbf{Y}) = (\tau\mathbf{X}, \mathbf{Y}), \qquad \tau f(\mathbf{X}) = f(\tau^{-1}\mathbf{X}).$$

In the literature, equivariance is primarily defined for a map $F : \mathbf{Z} \in \mathcal{X} \times \mathbb{R}^d \mapsto F_{\mathbf{Z}} \in \mathcal{F}(\mathcal{X}, \mathbb{R}^d)$ by:

$$\tau F_{\mathbf{Z}} = F_{\tau \mathbf{Z}}. \tag{S1}$$

Conversely, in the main text (Section 2), we defined equivariance for a prediction map through the representation function: a *prediction map* $\pi$ with representation function $r$ is $\mathcal{T}$-equivariant if and only if for all $\tau \in \mathcal{T}$,

$$r((\mathbf{X}, \mathbf{Y}), \mathbf{X}^*) = r((\tau\mathbf{X}, \mathbf{Y}), \tau\mathbf{X}^*). \tag{S2}$$

Eqs. S1 and S2 differ in that they deal with distinct objects. The former corresponds to the formal definition of an equivariant map, while the latter uses the representation function, which is simpler to work with for the scope of our paper.

To show the equivalence of the two definitions (Eqs. S1 and S2), we recall that a prediction map $\pi$ is defined through its representation $r$:

$$P_{\mathbf{Z}}(\mathbf{X}^*) = \pi(\cdot \mid \mathbf{Z}, \mathbf{X}^*) = p(\cdot \mid r((\mathbf{X}, \mathbf{Y}), \mathbf{X}^*)),$$

where $\mathbf{Z} \equiv (\mathbf{X}, \mathbf{Y})$. Then, starting from Eq. S1 applied to the prediction map,

$$P_{\tau(\mathbf{X},\mathbf{Y})} = \tau P_{(\mathbf{X},\mathbf{Y})} \Longleftrightarrow \forall \mathbf{X}^*, \, p\big(\cdot \mid r((\tau\mathbf{X}, \mathbf{Y}), \mathbf{X}^*)\big) = p\big(\cdot \mid r((\mathbf{X}, \mathbf{Y}), \tau^{-1}\mathbf{X}^*)\big)$$

$$\Longleftrightarrow \forall \mathbf{X}^*, \, r((\tau\mathbf{X}, \mathbf{Y}), \mathbf{X}^*) = r((\mathbf{X}, \mathbf{Y}), \tau^{-1}\mathbf{X}^*).$$

Noting that $\tau^{-1}r((\mathbf{X}, \mathbf{Y}), \mathbf{X}^*) = r((\mathbf{X}, \mathbf{Y}), \tau\mathbf{X}^*)$, by applying $\tau^{-1}$ to each side of the last line of the equation above, and swapping sides, we get:

$$P_{\tau(\mathbf{X},\mathbf{Y})} = \tau P_{(\mathbf{X},\mathbf{Y})} \Longleftrightarrow \forall \mathbf{X}^*, r((\mathbf{X}, \mathbf{Y}), \mathbf{X}^*) = r((\tau\mathbf{X}, \mathbf{Y}), \tau\mathbf{X}^*),$$

that is, Eqs. S1 and S2 are equivalent.

## A.2 Proof that RCNPs are equivariant

We recall from Eq. 3 in the main text that a prediction map $\pi$ with representation function $r$ is $\mathcal{T}$-*equivariant* with respect to a group $\mathcal{T}$ of transformations, $\tau : \mathcal{X} \to \mathcal{X}$, if and only if for all $\tau \in \mathcal{T}$:

$$r((\mathbf{X}, \mathbf{Y}), \mathbf{X}^\star) = r((\tau\mathbf{X}, \mathbf{Y}), \tau\mathbf{X}^\star),$$

where $\tau\mathbf{x} \equiv \tau(\mathbf{x})$ and $\tau\mathbf{X}$ is the set obtained by applying $\tau$ to all elements of $\mathbf{X}$.

Also recall that a prediction map $\pi$ and its representation function $r$ are *relational* with respect to a comparison function $g : \mathcal{X} \times \mathcal{X} \to \mathbb{R}^{d_{\text{comp}}}$ if and only if $r$ can be written solely through the set comparisons defined in Eq. 8 of the main text:

$$r((\mathbf{X}, \mathbf{Y}), \mathbf{X}^\star) = r\big(g((\mathbf{X}, \mathbf{Y}), (\mathbf{X}, \mathbf{Y})), g((\mathbf{X}, \mathbf{Y}), \mathbf{X}^\star), g(\mathbf{X}^\star, (\mathbf{X}, \mathbf{Y})), g(\mathbf{X}^\star, \mathbf{X}^\star)\big).$$

We can now prove the following Lemma from the main text.

**Lemma** (Lemma 4.4). *Let $\pi$ be a prediction map, $\mathcal{T}$ a transformation group, and $g$ a comparison function. If $\pi$ is relational with respect to $g$ and $g$ is $\mathcal{T}$-invariant, then $\pi$ is $\mathcal{T}$-equivariant.*

*Proof.* From Eq. 8 in the main text, if $g$ is $\mathcal{T}$-invariant, then for a target set $\mathbf{X}^\star$ and $\tau \in \mathcal{T}$:

$$g((\mathbf{X}, \mathbf{Y}), \mathbf{X}^\star) = \{(g(\mathbf{x}_n, \mathbf{x}_m^\star), \mathbf{y}_n)\} = \{(g(\tau\mathbf{x}_n, \tau\mathbf{x}_m^\star), \mathbf{y}_n)\} = g((\tau\mathbf{X}, \mathbf{Y}), \tau\mathbf{X}^\star),$$

for all $1 \le n \le N$ and $1 \le m \le M$. A similar equality holds for all the comparison sets in the definition of a relational prediction map (see above). Since $\pi$ is relational, and using the equality above, we can write its representation function $r$ as:

$$
\begin{aligned}
r((\mathbf{X}, \mathbf{Y}), \mathbf{X}^\star) &= r\big(g((\mathbf{X}, \mathbf{Y}), (\mathbf{X}, \mathbf{Y})), g((\mathbf{X}, \mathbf{Y}), \mathbf{X}^\star), g(\mathbf{X}^\star, (\mathbf{X}, \mathbf{Y})), g(\mathbf{X}^\star, \mathbf{X}^\star)\big) \\
&= r\big(g((\tau\mathbf{X}, \mathbf{Y}), (\tau\mathbf{X}, \mathbf{Y})), g((\tau\mathbf{X}, \mathbf{Y}), \tau\mathbf{X}^\star), g(\tau\mathbf{X}^\star, (\tau\mathbf{X}, \mathbf{Y})), g(\tau\mathbf{X}^\star, \tau\mathbf{X}^\star)\big) \\
&= r((\tau\mathbf{X}, \mathbf{Y}), \tau\mathbf{X}^\star).
\end{aligned}
$$

Thus, $\pi$ is $\mathcal{T}$-equivariant. $\qquad\qquad\square$

From the Lemma, we can prove the first main result of the paper, that RCNPs are equivariant prediction maps for a transformation group $\mathcal{T}$ provided a suitable comparison function $g$.

**Proposition** (Proposition 4.5). *Let $g$ be the comparison function used in a RCNP, and $\mathcal{T}$ a group of transformations. If $g$ is $\mathcal{T}$-invariant, the RCNP is $\mathcal{T}$-equivariant.*

*Proof.* In all RCNP models, elements of the context and target sets are always processed via a relational encoding. By definition (Eqs. 5 and 7 in the main text), the relational encoding is written solely as a function of $g((\mathbf{X}, \mathbf{Y}), (\mathbf{X}, \mathbf{Y}))$ and $g((\mathbf{X}, \mathbf{Y}), \mathbf{X}^\star)$, so the same holds for the representation function $r$. Thus, any RCNP is a relational prediction map with respect to $g$, and it follows that it is also a $\mathcal{T}$-equivariant prediction map according to Lemma 4.4. $\qquad\square$

## A.3 Proof that RCNPs are context-preserving

We prove here that RCNPs preserve information in the context set. We will do so by providing a construction for a function that reconstructs the context set given the relationally encoded vector $\boldsymbol{\rho}_m$ (up to transformations). As a disclaimer, the proofs in this section should be taken as 'existence proofs' but the provided construction is not practical. There are likely better context-preserving encodings, as shown by the empirical performance of RCNPs with moderately sized networks. Future theoretical work should provide stronger bounds on the size of the relational encoding, as recently established for DeepSets [43, 44].

For convenience, we recall Eq. 5 of the main text for the relational encoding provided a comparison function $g$:

$$\boldsymbol{\rho}_m = \rho_{\text{full}}(\mathbf{x}_m^\star, (\mathbf{X}, \mathbf{Y})) = \bigoplus_{n,n'=1}^{N} f_r\left(g(\mathbf{x}_n, \mathbf{x}_m^\star), \mathbf{R}_{nn'}\right), \qquad \mathbf{R}_{nn'} \equiv (g(\mathbf{x}_n, \mathbf{x}_{n'}), \mathbf{y}_n, \mathbf{y}_{n'}),$$

where $\mathbf{R} \equiv g((\mathbf{X}, \mathbf{Y}), g(\mathbf{X}, \mathbf{Y}))$ is the relational matrix. First, we show that from the relational encoding $\rho_{\text{full}}$ we can reconstruct the matrix $\mathbf{R}$, modulo permutations of rows or columns as defined below.

**Definition A.1.** Let $\mathbf{R}, \mathbf{R}' \in \mathbb{R}^{N \times N}$. The two matrices are *equal modulo permutations*, denoted $\mathbf{R} \cong \mathbf{R}'$, if there is a permutation $\sigma \in \text{Sym}(N)$ such that $\mathbf{R}_{nn'} = \mathbf{R}'_{\sigma(n)\sigma(n')}$, for all $1 \le n, n' \le N$.

In other words, $\mathbf{R} \cong \mathbf{R}'$ if $\mathbf{R}'$ is equal to $\mathbf{R}$ after an appropriate permutation of the indices of both rows and columns. Now we proceed with the reconstruction proof.

**Lemma A.2.** *Let $g : \mathcal{X} \times \mathcal{X} \to \mathbb{R}^{d_{\text{comp}}}$ be a comparison function, $(\mathbf{X}, \mathbf{Y})$ a context set, $\mathbf{X}^\star$ a target set, $\mathbf{R}$ the relational matrix and $\rho_{\text{full}}$ the relational encoding as per Eq. 5 in the main text. Then there is a reconstruction function $h$, for $\mathbf{r} = (\boldsymbol{\rho}_1, \ldots, \boldsymbol{\rho}_M)$, such that $h(\mathbf{r}) = \mathbf{R}'$ where $\mathbf{R}' \cong \mathbf{R}$.*

*Proof.* For this proof, it is sufficient to show that there is a function $h$ applied to a local representation of any context point, $\boldsymbol{\rho}_m = \rho_{\text{full}}(\mathbf{x}_m^\star, (\mathbf{X}, \mathbf{Y}))$, such that $h(\boldsymbol{\rho}_m) = \mathbf{R}'$, with $\mathbf{R}' \cong \mathbf{R}$. In this proof, we also consider that all numbers in a computer are represented up to numerical precision, so with $\mathbb{R}$ we denote the set of real numbers physically representable in a chosen floating point representation. As a final assumption, without loss of generality, we take $\mathbf{y}_n \ne \mathbf{y}_{n'}$ for $n \ne n'$. If there are non-unique elements $\mathbf{y}_n$ in the context set, in practice we can always disambiguate them by adding a small jitter.

Let $f_r : \mathcal{R} \to \mathbb{R}^{d_{\text{rel}}}$ be the relational encoder of Eq. 5 of the main text, a neural network parametrized by $\boldsymbol{\theta}_r$, where $\mathcal{R} = \mathbb{R}^{d_{\text{comp}}} \times \mathbb{R}^{d_{\text{comp}}+2d_y}$. Here, we also include in $\boldsymbol{\theta}_r$ hyperparameters defining the network architecture, such as the number of layers and number of hidden units per layer. We pick $d_{\text{rel}}$ large enough such that we can build a one-hot encoding of the elements of $\mathcal{R}$, i.e., there is an injective mapping from $\mathcal{R}$ to $\{0, 1\}^{d_{\text{rel}}}$ such that one and only one element of the output vector is 1. This injective mapping exists, since $\mathcal{R}$ is discrete due to finite numerical precision. We select $\boldsymbol{\theta}_r$ such that $f_r$ approximates the one-hot encoding up to arbitrary precision, which is achievable thanks to universal approximation theorems for neural networks [29]. Thus, $\boldsymbol{\rho}_m = \bigoplus_{n,n'} f_r(g(\mathbf{x}_n, \mathbf{x}_m^\star), \mathbf{R}_{nn'})$ is a (astronomically large) vector with ones at the elements that denote $(g(\mathbf{x}_n, \mathbf{x}_m^\star), \mathbf{R}_{nn'})$ for $1 \le n, n' \le N$.

Finally, we can build a reconstruction function $h$ that reads out $\boldsymbol{\rho}_m$, the sum of one-hot encoded vectors, mapping each 'one', based on its location, back to the corresponding input $(g(\mathbf{x}_n, \mathbf{x}_m^\star), \mathbf{R}_{nn'})$. We have then all the information to build a matrix $\mathbf{R}'$ indexed by $\mathbf{y}$, that is

$$\mathbf{R}'_{\mathbf{y}\mathbf{y}'} = \mathbf{R}_{nn'}, \qquad \text{with } \mathbf{y} = \mathbf{y}_n, \mathbf{y}' = \mathbf{y}_{n'}, \quad \text{for some } \mathbf{y}, \mathbf{y}' \in \mathbf{Y}, 1 \le n, n' \le N.$$

Since we assumed $\mathbf{y}_n \ne \mathbf{y}_{n'}$ for $n \ne n'$, $\mathbf{R}'$ is unique up to permutations of the rows and columns, and by construction $\mathbf{R}' \cong \mathbf{R}$. $\qquad \square$

We now define the matrix

$$\mathbf{Q}_{mnn'} = (g(\mathbf{x}_n, \mathbf{x}_m^\star), g(\mathbf{x}_n, \mathbf{x}_{n'}), \mathbf{y}_n, \mathbf{y}_{n'}) = (g(\mathbf{x}_n, \mathbf{x}_m^\star), \mathbf{R}_{nn'}),$$

for $1 \le m \le M$ and $1 \le n, n' \le N$. We recall that a comparison function $g$ is *context-preserving* with respect to a transformation group $\mathcal{T}$ if for any context set $(\mathbf{X}, \mathbf{Y})$ and target set $\mathbf{X}^\star$, for all $1 \le m \le M$, there is a submatrix $\mathbf{Q}' \subseteq \mathbf{Q}_{m::}$, a reconstruction function $\gamma$, and a transformation $\tau \in \mathcal{T}$ such that $\gamma(\mathbf{Q}') = (\tau \mathbf{X}, \mathbf{Y})$. We showed in Section 4.2 of the main text that the distance comparison function $g_{\text{dist}}$ is context-preserving with respect to the group of rigid transformations; and that the difference comparison function $g_{\text{diff}}$ is context-preserving with respect to the translation group. Similarly, a family of functions $h_{\boldsymbol{\theta}}((\mathbf{X}, \mathbf{Y}), \mathbf{X}^\star) \to \mathbb{R}^{d_{\text{rep}}}$ is *context-preserving* under $\mathcal{T}$ if there exists $\boldsymbol{\theta} \in \boldsymbol{\Theta}$, $d_{\text{rep}} \in \mathbb{N}$, a *reconstruction function* $\gamma$, and a transformation $\tau \in \mathcal{T}$ such that $\gamma(h_{\boldsymbol{\theta}}((\mathbf{X}, \mathbf{Y}), \mathbf{X}^\star)) = (\tau \mathbf{X}, \mathbf{Y})$.

With the construction and definitions above, we can now show that RCNPs are in principle context-preserving. In particular, we will apply the definition of context-preserving functions $h$ to the class of representation functions $r$ of the RCNP models introduced in the paper, for which $d_{\text{rep}} = M d_{\text{rel}}$ since $\mathbf{r} = (\boldsymbol{\rho}_1, \ldots, \boldsymbol{\rho}_M)$. The proof applies both to the full RCNP and the simple (diagonal) RCNP, for an appropriate choice of the comparison function $g$.

**Proposition** (Proposition 4.8). *Let $\mathcal{T}$ be a transformation group and $g$ the comparison function used in a FullRCNP. If $g$ is context-preserving with respect to $\mathcal{T}$, then the representation function $r$ of the FullRCNP is context-preserving with respect to $\mathcal{T}$.*

*Proof.* Given Lemma A.2, for any given $m$, we can reconstruct $\mathbf{Q}_{m::} = (\ldots, \mathbf{R})$ (modulo permutations) from the relational encoding. Since $g$ is context-preserving with respect to $\mathcal{T}$, we can

reconstruct $(\mathbf{X}, \mathbf{Y})$ from $\mathbf{R}$ modulo a transformation $\tau \in \mathcal{T}$. Thus, $\rho_{\text{full}}$ is context-preserving with respect to $\mathcal{T}$ and so is $r$. □

**Proposition** (Proposition 4.9). *Let $\mathcal{T}$ be the translation group and $g_{\text{diff}}$ the difference comparison function. The representation of the simple RCNP model with $g_{\text{diff}}$ is context-preserving with respect to $\mathcal{T}$.*

*Proof.* As shown in the main text, the difference comparison function is context-preserving for $\mathbf{Q}'_{m:} = \mathbf{Q}_{mnn}|_{n=1}^{N} = (g(\mathbf{x}_n, \mathbf{x}^\star_m), \mathbf{R}_{nn})_{n=1}^{N}$, for a given $m$. In other words, we can reconstruct $(\mathbf{X}, \mathbf{Y})$ given only the diagonal of the $\mathbf{R}$ matrix and the comparison between a single target input $\mathbf{x}^\star_m$ and the elements of the context set, $g(\mathbf{x}_n, \mathbf{x}^\star_m)$. Following a construction similar to the one in Lemma A.2, we can map $\boldsymbol{\rho}_m$ back to $\mathbf{Q}'_{m:}$. Then since $g$ is context-preserving with respect to $\mathbf{Q}'_{m:}$, we can reconstruct $(\mathbf{X}, \mathbf{Y})$ up to translations. □

A potentially surprising consequence of our context-preservation theorems is that in principle no information is lost about the entire context set. Thus, *in principle* an RCNP with a sufficiently large network is able to reconstruct any high-order interaction of the context set, even though the RCNP construction only builds upon interactions of pairs of points. However, higher-order interactions might be harder to encode *in practice*. Since the RCNP representation is built on two-point interactions, depending on the network size, it may be harder for the network to effectively encode the simultaneous interaction of many context points.

## B  Methods

### B.1  Training procedure, error bars and statistical significance

This section describes the training and evaluation procedures used in our regression experiments. The training procedure follows [3, Appendix G]. Stochastic gradient descent is performed via the Adam algorithm [21] with a learning rate specified in each experiment. After each epoch during model training, each model undergoes validation using a pre-generated validation set. The validation score is a confidence bound based on the log-likelihood values. Specifically, the mean ($\mu_{\text{val}}$) and standard deviation ($\sigma_{\text{val}}$) of the log-likelihood values over the entire validation set are used to calculate the validation score as $\mu_{\text{val}} - 1.96\sigma_{\text{val}}/\sqrt{N_{\text{val}}}$ where $N_{\text{val}}$ is the validation dataset size. The validation score is compared to the previous best score observed within the training run, and if the current validation score is higher, the current model parameters (e.g., weights and biases for neural networks) are saved. The models are trained for a set number of epochs in each experiment, and when the training is over, the model is returned with the parameters that resulted in the highest validation score.

To ensure reliable statistical analysis and fair comparisons, we repeat model training using the above procedure ten times in each experiment, with each run utilizing different seeds for training and validation dataset generation and model training. In practice, we generated ten seed sets so that each training run with the same model utilizes different seeds, but we maintain consistency by using the same seeds across different models in the same experiment. Each model is then represented with ten training outcomes in each experiment, and we evaluate all training outcomes with a fixed evaluation set or evaluation sets.

We evaluate the methods based on two performance measures: *(i)* the log-likelihood, normalized with respect to the number of target points, and *(ii)* the Kullback-Leibler (KL) divergence between the posterior prediction map $\pi$ and a target $\pi^*$, defined as

$$\text{KL}(\pi||\pi^*) = \int \pi(x) \log \frac{\pi(x)}{\pi^*(x)} \mathrm{d}x,$$

assuming that $\pi$ is absolutely continuous to $\pi^*$ with respect to a measure $\lambda$ (usually, the Lebesgue measure). We report the KL normalized with respect to the number of target points.

Throughout our experiments, we rely on the KL divergence to evaluate the experiments with Gaussian process (GP) generated functions ('Gaussian'), for which we can compute the ground-truth target $\pi^*$ (the multivariate normal prediction of the posterior GP). We use the normalized log-likelihood for all the other ('non-Gaussian') cases. For each model and evaluation set, we report both the average score

calculated across training runs and the standard deviation between scores observed in each training run.

We compare models based on the average scores and also run pairwise statistical tests to identify models that resulted in comparable performance. The results from pairwise comparisons are used throughout the paper in the results tables to highlight in bold the models that are considered best in each experiment. Specifically, we use the following method to determine the best models:

- First, we highlight in bold the model (A) that has the best empirical mean metric.

- Then, for each alternative model (B), we run a one-sided Student's t-test for paired samples, with the alternative hypothesis that model (B) has a higher (better) mean than model (A), and the null hypothesis that the two models have the same mean. Samples are paired since they share a random seed (i.e., the same training data).

- The models that do *not* lead to statistically significant ($p < 0.05$) rejection of the null hypothesis are highlighted in bold as well.

### B.2 Experimental details and reproducibility

The experiments carried out in this work used the open-source neural processes package released with previous work [3]. The package is distributed under the MIT license and available at `https://github.com/wesselb/neuralprocesses` [1]. We extended the package with the proposed relational model architectures and the new experiments considered in this work. Our implementation of RCNP is available at `https://github.com/acerbilab/relational-neural-processes`.

For the purpose of open science and reproducibility, all training and evaluation details are provided in the following sections of this Appendix as well as in the Github repository linked above.

### B.3 Details on Figure 1

In Figure 1 in the main text, we compare CNP and RCNP models. The RCNP model used in this example utilizes a stationary kernel to encode translation equivariance. The encoding in the CNP model and the relational encoding in the RCNP model are 256-dimensional, and both models used an encoder network with three hidden layers and 256 hidden units per layer. In addition, both models used a decoder network with six hidden layers and 256 hidden units per layer. We used the CNP architecture from previous work [3] and modeled the RCNP architecture on the CNP architecture. The encoder and decoder networks use ReLU activation functions.

We trained the neural process models with datasets generated based on random functions sampled from a noiseless Gaussian process prior with a Matérn-$\frac{5}{2}$ covariance function with lengthscale $\ell = 0.5$. The datasets were generated by sampling 1–30 context points and 50 target points from interval $[-2, 2]$. The neural process models were trained for 20 epochs with $2^{14}$ datasets in each epoch and learning rate $3 \cdot 10^{-4}$. The datasets used in training the CNP and RCNP models were generated with the same seed.

Figure 1 shows model predictions in two regression tasks. First, we visualized the predicted mean and credible bounds calculated as 1.96 times the predicted standard deviation in the range $[-2, 2]$ given 10 context points. This corresponds to the standard *interpolation (INT)* task. Then we shifted the context points and visualized the predicted mean and credible bounds in the range $[2, 6]$. This corresponds to an *out-of-input-distribution (OOID)* task. The Gaussian process model used as a reference had Matérn-$\frac{5}{2}$ covariance function with lengthscale $\ell = 0.5$, the same process used to train the CNP and RCNP models.

## C Computational cost analysis

### C.1 Inference time analysis

In this section, we provide a quantitative analysis of the computational cost associated with running various models, reporting the inference time under different input dimensions $d_x$ and varying context set sizes $N$. We compare our simple and full RCNP models with the CNP and ConvCNP families.

All results are calculated on an Intel Core i7-12700K CPU, under the assumption that these models can be deployed on devices or local machines without GPU access.

Table S1 shows the results with different input dimensions $d_x$. We set the size of both context and target sets to 20. Firstly, we can see the runtime of convolutional models increase significantly as they involve costly convolutional operations, and cannot be applied in practice to $d_x > 2$. Conversely, the runtime cost remains approximately constant across $d_x$ for the other models. Secondly, our RCNP models have inference time close to that of CNP models, while FullRCNP models are slower (although still constant in terms of $d_x$). Since context set size is the main factor affecting the inference speed of our RCNP models, we delve further into the computational costs associated with varying context set sizes $N$, keeping the input dimension $d_x = 1$ and target set size $M = 20$ constant (Table S2). Given that RCNP has a cost of $O(NM)$, we observe a small, steady increase in runtime. Conversely, the FullRCNP models, with their $O(N^2M)$ cost, show an approximately quadratic surge in runtime as $N$ increases, as expected.

Table S1: Comparison of computational costs across different models under different input dimensions $d_x$. The size of both context sets ($N$) and target sets ($M$) is set to 20. We report the mean value and (standard deviation) derived from 50 forward passes executed on various randomly generated datasets.

| | Runtime ($\times 10^{-3}$ s) | | | |
| --- | --- | --- | --- | --- |
| | $d_x = 1$ | $d_x = 2$ | $d_x = 3$ | $d_x = 5$ |
| RCNP | 2.35 (0.39) | 2.29 (0.28) | 2.28 (0.32) | 2.25 (0.27) |
| RGNP | 2.31 (0.30) | 2.41 (0.34) | 2.31 (0.26) | 2.50 (0.40) |
| FullRCNP | 8.24 (0.46) | 9.09 (1.78) | 8.27 (0.56) | 9.08 (1.72) |
| FullRGNP | 8.40 (0.56) | 9.25 (1.85) | 8.37 (0.62) | 9.20 (1.76) |
| ConvCNP | 4.46 (0.38) | 31.19 (2.43) | - | - |
| ConvGNP | 4.51 (0.48) | 38.04 (2.31) | - | - |
| CNP | 1.86 (0.30) | 2.10 (0.38) | 2.14 (0.30) | 2.09 (0.29) |
| GNP | 2.01 (0.27) | 2.09 (0.26) | 2.20 (0.26) | 2.10 (0.32) |

Table S2: Comparison of computational costs across different models under different context set sizes $N$. We maintain the input dimension at $d_x = 1$, and a fixed target set size of $M = 20$. We report the mean value and (standard deviation) derived from 50 forward passes executed on various randomly generated datasets.

| | Runtime ($\times 10^{-3}$ s) | | | |
| --- | --- | --- | --- | --- |
| | $N = 10$ | $N = 20$ | $N = 50$ | $N = 100$ |
| RCNP | 1.90 (0.26) | 2.34 (0.48) | 2.40 (0.30) | 2.90 (0.46) |
| RGNP | 2.10 (0.26) | 2.53 (0.34) | 2.55 (0.22) | 2.93 (0.37) |
| FullRCNP | 2.83 (0.33) | 8.80 (1.59) | 38.03 (4.71) | 154.17 (16.04) |
| FullRGNP | 3.08 (0.34) | 8.68 (1.02) | 38.33 (5.88) | 161.07 (19.54) |
| ConvCNP | 4.47 (0.43) | 4.47 (0.51) | 5.10 (0.62) | 5.83 (0.88) |
| ConvGNP | 5.10 (0.54) | 4.55 (0.35) | 5.26 (0.50) | 4.70 (0.46) |
| CNP | 1.95 (0.27) | 1.86 (0.28) | 1.87 (0.21) | 1.97 (0.19) |
| GNP | 2.48 (0.70) | 2.03 (0.19) | 2.17 (0.48) | 2.11 (0.23) |

## C.2 Overall training time

For the entire paper, we conducted all experiments, including baseline model computations and preliminary experiments not included in the paper, on a GPU cluster consisting of a mix of Tesla P100, Tesla V100, and Tesla A100 GPUs. We roughly estimate the total computational consumption to be around 25 000 GPU hours. A more detailed evaluation of computing for each set of experiments is reported when available in the following sections.

## D Details of synthetic regression experiments

We report details and additional results for the synthetic regression experiments from Section 5.1 of the main text. The experiments compare neural process models trained on data sampled from both Gaussian and non-Gaussian synthetic functions, where 'Gaussian' refers to functions sampled from a

Gaussian process (GP). Our procedure follows closely the synthetic regression experiments presented in [3].

## D.1 Models

We compare our RCNP models with the CNP and ConvCNP model families in regression tasks with input dimensions $d_x = \{1, 2, 3, 5, 10\}$. The CNP and GNP models used in this experiment encode the context sets as 256-dimensional vectors when $d_x < 5$ and as 128-dimensional vectors when $d_x \geq 5$. Similarly, all relational models including RCNP, RGNP, FullRCNP, and FullRGNP produce relational encodings with dimension 256 when $d_x < 5$ and dimension 128 when $d_x \geq 5$. The encoding network used in both model families to produce the encoding or relational encoding is a three-layer MLP, featuring 256 hidden units per layer for $d_x < 5$ and 128 for $d_x \geq 5$. We also maintain the same setting across all CNP and RCNP models in terms of the decoder network architecture, using a six-layer MLP with 256 hidden units per layer for $d_x < 5$ and 128 for $d_x \geq 5$. The encoder and decoder networks use ReLU activation functions. Finally, the convolutional models ConvCNP and ConvGNP, which are included in experiments where $d_x = \{1, 2\}$, are employed with the configuration detailed in [3, Appendix F], and GNP, RGNP, FullRGNP, and ConvGNP models all use `linear` covariance with 64 basis functions.

Neural process models are trained with datasets representing a regression task with context and target features. The datasets used in these experiments were generated based on random functions sampled from Gaussian and non-Gaussian stochastic processes. The models were trained for 100 epochs with $2^{14}$ datasets in each epoch and learning rate $3 \cdot 10^{-4}$. The validation sets used in training included $2^{12}$ datasets and the evaluation sets used to compare the models in interpolation (INT) and out-of-input-distribution (OOID) tasks included $2^{12}$ datasets each.

## D.2 Data

We used the following functions to generate synthetic Gaussian and non-Gaussian data:

- **Exponentiated quadratic (EQ).** We sample data from a GP with an EQ covariance function:

$$f \sim \mathcal{GP}\left(\mathbf{0}, \exp\left(-\frac{||\mathbf{x} - \mathbf{x}'||_2^2}{2\ell^2}\right)\right), \tag{S3}$$

  where $\ell$ is the lengthscale.

- **Matérn-$\frac{5}{2}$.** We sample data from a GP with a Matérn-$\frac{5}{2}$ covariance function:

$$f \sim \mathcal{GP}\left(\mathbf{0}, \left(1 + \sqrt{5}r + \frac{5}{3}r^2\right)\exp\left(-\sqrt{5}r\right)\right), \tag{S4}$$

  where $r = \frac{||\mathbf{x} - \mathbf{x}'||_2}{\ell}$ and $\ell$ is the lengthscale.

- **Weakly periodic.** We sample data from a GP with a weakly periodic covariance function:

$$f \sim \mathcal{GP}\left(\mathbf{0}, \exp\left(-\frac{1}{2\ell_d^2}||\mathbf{x} - \mathbf{x}'||_2^2 - \frac{2}{\ell_p^2}\left|\left|\sin\left(\frac{\pi}{p}(\mathbf{x} - \mathbf{x}')\right)\right|\right|_2^2\right)\right), \tag{S5}$$

  where $\ell_d$ is the lengthscale that decides how quickly the similarity between points in the output of the function decays as their inputs move apart, $\ell_p$ determines the lengthscale of periodic variations, and $p$ denotes the period.

- **Sawtooth.** We sample data from a sawtooth function characterized by a stochastic frequency, orientation, and phase as presented by:

$$f \sim \omega\langle\mathbf{x}, \mathbf{u}\rangle_2 + \phi \mod 1, \tag{S6}$$

  where $\omega \sim \mathcal{U}(\Omega)$ is the frequency of the sawtooth wave, the direction of the wave is given as $\mathbf{u} \sim \mathcal{U}(\{\mathbf{x} \in \mathbb{R}^{d_x} : ||\mathbf{x}||_2 = 1\})$, and $\phi \sim \mathcal{U}(0, 1)$ determines the phase.

- **Mixture.** We sample data randomly from either one of the three GPs or the sawtooth process, with each having an equal probability to be chosen.

In this set of experiments, we evaluate the models using varying input dimensions $d_x = \{1, 2, 3, 5, 10\}$. To maintain a roughly equal level of difficulty across data with varying input dimensions $d_x$, the hyperparameters for the above data generation processes are selected in accordance with $d_x$:

$$\ell = \sqrt{d_x}, \quad \ell_d = 2 \cdot \sqrt{d_x}, \quad \ell_p = 4 \cdot \sqrt{d_x}, \quad p = \sqrt{d_x}, \quad \Omega = \left[\frac{1}{2\sqrt{d_x}}, \frac{1}{\sqrt{d_x}}\right]. \quad \text{(S7)}$$

For the EQ, Matérn-$\frac{5}{2}$, and weakly periodic functions, we additionally add independent Gaussian noise with $\sigma^2 = 0.05$.

The datasets representing regression tasks were generated by sampling context and target points from the synthetic data as follows. The number of context points sampled from each EQ, Matérn-$\frac{5}{2}$, or weakly periodic function varied uniformly between 1 and $30 \cdot d_x$, and the number of target points was fixed to $50 \cdot d_x$. Since the datasets sampled from the sawtooth and mixture functions represent more difficult regression problems, we sampled these functions for 1–30 context points when $d_x = 1$ and $1$–$50 \cdot d_x$ context points when $d_x > 1$, and we also fixed the number of target points to $100 \cdot d_x$.

All training and validation datasets were sampled from the range $\mathcal{X} = [-2, 2]^{d_x}$ while evaluation sets were sampled in two ways. To evaluate the models in an *interpolation* (INT) task, we generated evaluation datasets by sampling context and target points from the same range that was used during training, $\mathcal{X}_{\text{test}} = [-2, 2]^{d_x}$. To evaluate the models in an *out-of-input-distribution* (OOID) task, we generated evaluation datasets with context and target points sampled from a range that is outside the boundaries of the training range, specifically $\mathcal{X}_{\text{test}} = [2, 6]^{d_x}$.

## D.3  Full results

The results presented in the main text (Section 5.1) compared the proposed RCNP and RGNP models to baseline and convolutional CNP and GNP models in selected synthetic regression tasks. The full results encompassing a wider selection of tasks and an extended set of models are presented in Table S3, S4, S5, S6, S7. We constrained the experiments with EQ and Matérn-$\frac{5}{2}$ tasks (Table S3 and S4) to input dimensions $d_x = \{1, 2, 3, 5\}$ owing to the prohibitive computational memory costs associated with the FullRCNP model included in these experiments. The other experiments (Table S5–S7) were repeated with input dimensions $d_x = \{1, 2, 3, 5, 10\}$. Our RCNP models demonstrate consistently competitive performance when compared with ConvCNP models in scenarios with low dimensional data, and significantly outperform the CNP family of models when dealing with data of higher dimensions.

Table S3: **Synthetic (EQ).** Comparison of the *interpolation* (INT) and *out-of-input-distribution* (OOID) performance of our RCNP models with different CNP baselines on EQ function with varying input dimensions. We use normalized Kullback-Leibler divergences as our metric and show mean and (standard deviation) obtained from 10 runs with different seeds. *Trivial* refers to a model that predicts a Gaussian distribution utilizing the empirical mean and standard deviation derived from the context outputs. "F" denotes failed attempts that yielded very bad results. Missing entries could not be run. Statistically significantly best results are **bolded**.

| | $d_x = 1$ | | $d_x = 2$ | | $d_x = 3$ | | $d_x = 5$ | |
| --- | --- | --- | --- | --- | --- | --- | --- | --- |
| | INT | OOID | INT | OOID | INT | OOID | INT | OOID |
| RCNP (sta) | 0.22 (0.00) | 0.22 (0.00) | 0.26 (0.00) | 0.26 (0.00) | 0.40 (0.01) | 0.40 (0.01) | 0.45 (0.00) | 0.45 (0.00) |
| RGNP (sta) | 0.01 (0.00) | 0.01 (0.00) | 0.03 (0.00) | 0.03 (0.00) | **0.05** (0.00) | **0.05** (0.00) | **0.11** (0.00) | **0.11** (0.00) |
| FullRCNP (iso) | 0.22 (0.00) | 0.22 (0.00) | 0.26 (0.00) | 0.26 (0.00) | 0.31 (0.00) | 0.31 (0.00) | 0.35 (0.00) | 0.35 (0.00) |
| FullRGNP (iso) | 0.03 (0.00) | 0.03 (0.00) | 0.08 (0.00) | 0.08 (0.00) | 0.14 (0.00) | 0.14 (0.00) | 0.25 (0.00) | 0.25 (0.00) |
| ConvCNP | 0.21 (0.00) | 0.21 (0.00) | 0.22 (0.00) | 0.22 (0.00) | - | - | - | - |
| ConvGNP | **0.00** (0.00) | **0.00** (0.00) | **0.01** (0.00) | **0.01** (0.00) | - | - | - | - |
| CNP | 0.25 (0.00) | 2.21 (0.70) | 0.33 (0.00) | 4.54 (1.76) | 0.44 (0.00) | 3.30 (1.55) | 0.57 (0.00) | 1.22 (0.09) |
| GNP | 0.02 (0.00) | F | 0.05 (0.00) | 2.25 (0.61) | 0.09 (0.01) | 2.54 (1.44) | 0.19 (0.00) | 0.74 (0.02) |
| *Trivial* | 1.03 (0.00) | 1.03 (0.00) | 1.13 (0.00) | 1.13 (0.00) | 1.12 (0.00) | 1.12 (0.00) | 1.03 (0.00) | 1.03 (0.00) |

We also conducted an additional experiment to explore all the possible combinations of comparison functions with the RCNP and FullRCNP models on EQ and Matérn-$\frac{5}{2}$ tasks. Table S8 provides a comprehensive view of the results for EQ tasks across three input dimensions. The empirical results support our previous demonstration that RCNP models are capable of incorporating translation-equivariance with no loss of information: the stationary versions of the RCNP model

Table S4: **Synthetic (Matérn-$\frac{5}{2}$).** Comparison of the *interpolation* (INT) and *out-of-input-distribution* (OOID) performance of our RCNP models with different CNP baselines on Matérn-$\frac{5}{2}$ function with varying input dimensions. We use normalized Kullback-Leibler divergences as our metric and show mean and (standard deviation) obtained from 10 runs with different seeds. *Trivial* refers to a model that predicts a Gaussian distribution utilizing the empirical mean and standard deviation derived from the context outputs. "F" denotes failed attempts that yielded very bad results. Missing entries could not be run. Statistically significantly best results are **bolded**.

| | $d_x = 1$ | | $d_x = 2$ | | $d_x = 3$ | | $d_x = 5$ | |
|---|---|---|---|---|---|---|---|---|
| | INT | OOID | INT | OOID | INT | OOID | INT | OOID |
| RCNP (sta) | 0.25 (0.00) | 0.25 (0.00) | 0.30 (0.00) | 0.30 (0.00) | 0.39 (0.00) | 0.39 (0.00) | 0.35 (0.00) | 0.35 (0.00) |
| RGNP (sta) | 0.01 (0.00) | 0.01 (0.00) | 0.03 (0.00) | 0.03 (0.00) | **0.05** (0.00) | **0.05** (0.00) | **0.11** (0.00) | **0.11** (0.00) |
| FullRCNP (iso) | 0.25 (0.00) | 0.25 (0.00) | 0.30 (0.00) | 0.30 (0.00) | 0.32 (0.00) | 0.32 (0.00) | 0.29 (0.00) | 0.29 (0.00) |
| FullRGNP (iso) | 0.03 (0.00) | 0.03 (0.00) | 0.09 (0.00) | 0.09 (0.00) | 0.16 (0.00) | 0.16 (0.00) | 0.21 (0.00) | 0.21 (0.00) |
| ConvCNP | 0.24 (0.00) | 0.24 (0.00) | 0.26 (0.00) | 0.26 (0.00) | - | - | - | - |
| ConvGNP | **0.00** (0.00) | **0.00** (0.00) | **0.01** (0.00) | **0.01** (0.00) | - | - | - | - |
| CNP | 0.29 (0.00) | F | 0.39 (0.00) | 6.75 (2.72) | 0.46 (0.00) | 1.75 (0.42) | 0.47 (0.00) | 0.93 (0.02) |
| GNP | 0.02 (0.00) | 2.96 (1.77) | 0.07 (0.00) | 1.86 (0.26) | 0.11 (0.00) | 1.23 (0.17) | 0.19 (0.00) | 0.62 (0.02) |
| *Trivial* | 1.04 (0.00) | 1.04 (0.00) | 1.06 (0.00) | 1.06 (0.00) | 0.98 (0.00) | 0.98 (0.00) | 0.79 (0.00) | 0.79 (0.00) |

Table S5: **Synthetic (weakly-periodic).** Comparison of the *interpolation* (INT) and *out-of-input-distribution* (OOID) performance of our RCNP models with different CNP baselines on weakly-periodic function with varying input dimensions. We use normalized Kullback-Leibler divergences as our metric and show mean and (standard deviation) obtained from 10 runs with different seeds. *Trivial* refers to a model that predicts a Gaussian distribution utilizing the empirical mean and standard deviation derived from the context outputs. "F" denotes failed attempts that yielded very bad results. Missing entries could not be run. Statistically significantly best results are **bolded**.

| | $d_x = 1$ | | $d_x = 2$ | | $d_x = 3$ | | $d_x = 5$ | | $d_x = 10$ | |
|---|---|---|---|---|---|---|---|---|---|---|
| | INT | OOID | INT | OOID | INT | OOID | INT | OOID | INT | OOID |
| RCNP (sta) | 0.24 (0.00) | 0.24 (0.00) | 0.24 (0.00) | 0.24 (0.00) | 0.28 (0.00) | 0.28 (0.00) | 0.31 (0.00) | 0.31 (0.00) | 0.31 (0.00) | 0.31 (0.00) |
| RGNP (sta) | 0.03 (0.00) | 0.03 (0.00) | 0.05 (0.00) | 0.05 (0.00) | **0.05** (0.00) | **0.05** (0.00) | **0.08** (0.00) | **0.08** (0.00) | **0.11** (0.00) | **0.11** (0.00) |
| ConvCNP | 0.21 (0.00) | 0.21 (0.00) | 0.20 (0.00) | 0.20 (0.00) | - | - | - | - | - | - |
| ConvGNP | **0.01** (0.00) | **0.01** (0.00) | **0.02** (0.00) | **0.02** (0.00) | - | - | - | - | - | - |
| CNP | 0.31 (0.00) | 2.88 (0.91) | 0.39 (0.00) | 1.81 (0.43) | 0.39 (0.00) | 1.58 (0.50) | 0.42 (0.00) | 2.20 (0.81) | 0.75 (0.00) | 1.03 (0.11) |
| GNP | 0.06 (0.00) | F | 0.07 (0.00) | 2.57 (0.76) | 0.08 (0.01) | 1.47 (0.27) | 0.11 (0.01) | 0.62 (0.04) | 0.22 (0.01) | 0.49 (0.05) |
| *Trivial* | 0.78 (0.00) | 0.78 (0.00) | 0.81 (0.00) | 0.81 (0.00) | 0.80 (0.00) | 0.80 (0.00) | 0.77 (0.00) | 0.77 (0.00) | 0.76 (0.00) | 0.76 (0.00) |

deliver performance comparable to the stationary FullRCNP models. Conversely, since the RCNP model is not context-preserving for rigid transformations, the isotropic versions of the RCNP model exhibit inferior performance in comparison to the isotropic FullRCNPs.

# E  Details of Bayesian optimization experiments

We report here details and additional results for the Bayesian optimization experiments from Section 5.2 of the main text. We evaluated our proposed approach on a common synthetic optimization test function, the Hartmann function in its three and six dimensional versions [40, p.185].[1] Each of two test functions is evaluated on a $d_x$-dimensional hypercube $\mathcal{X} = [0,1]^{d_x}$.

## E.1  Models

We compared our proposals, RCNP and RGNP, against CNP and GNP as well as their attentive variants ACNP and AGNP. A GP with a Matérn-$\frac{5}{2}$ kernel additionally served as the "gold standard" to be compared against. We mostly followed the architectural choices discussed in Section D.1. However, we kept the number of hidden units per layer fixed at 256 and the relational encoding dimension at 128, irrespective of the dimensionality $d_x$. The models were trained with a learning rate of $3 \cdot 10^{-4}$ over up to 500 epochs with $2^{14}$ datasets in each epoch. A validation set with $2^{12}$ datasets was used to track the training progress and early stopping was performed if the validation score had not improved for 150 epochs.

---

[1]See https://www.sfu.ca/~ssurjano/optimization.html for their precise definitions and minima.

Table S6: **Synthetic (sawtooth).** Comparison of the *interpolation* (INT) and *out-of-input-distribution* (OOID) performance of our RCNP models with different CNP baselines on sawtooth function with varying input dimensions. We use normalized log-likelihoods as our metric and show mean and (standard deviation) obtained from 10 runs with different seeds. *Trivial* refers to a model that predicts a Gaussian distribution utilizing the empirical mean and standard deviation derived from the context outputs. "F" denotes failed attempts that yielded very bad results. Missing entries could not be run. Statistically significantly best results are **bolded**.

| | $d_x = 1$ | | $d_x = 2$ | | $d_x = 3$ | | $d_x = 5$ | | $d_x = 10$ | |
|---|---|---|---|---|---|---|---|---|---|---|
| | INT | OOID | INT | OOID | INT | OOID | INT | OOID | INT | OOID |
| RCNP (sta) | 3.03 (0.06) | 3.04 (0.06) | 1.73 (0.03) | 1.74 (0.03) | 0.85 (0.01) | 0.85 (0.01) | 0.44 (0.00) | 0.44 (0.00) | 0.75 (0.00) | 0.75 (0.00) |
| RGNP (sta) | 3.90 (0.09) | 3.90 (0.10) | 2.13 (0.33) | 2.13 (0.32) | **1.09** (0.01) | **1.09** (0.01) | **1.13** (0.05) | **1.13** (0.05) | **1.33** (0.03) | **1.32** (0.03) |
| ConvCNP | 3.64 (0.04) | 3.64 (0.04) | 3.66 (0.04) | 3.66 (0.04) | - | - | - | - | - | - |
| ConvGNP | **3.94** (0.11) | **3.97** (0.08) | **4.11** (0.03) | 3.99 (0.11) | - | - | - | - | - | - |
| CNP | 2.25 (0.02) | F | 1.15 (0.45) | -3.27 (4.72) | 0.36 (0.28) | -0.37 (0.12) | -0.03 (0.10) | -0.22 (0.03) | 0.27 (0.00) | -2.29 (0.67) |
| GNP | 0.83 (0.04) | F | 1.04 (0.09) | F | 0.23 (0.13) | F | 0.02 (0.05) | F | 0.03 (0.04) | F |
| *Trivial* | -0.27 (0.00) | -0.27 (0.00) | -0.19 (0.00) | -0.19 (0.00) | -0.19 (0.00) | -0.19 (0.00) | -0.18 (0.00) | -0.18 (0.00) | -0.14 (0.00) | -0.14 (0.00) |

Table S7: **Synthetic (mixture).** Comparison of the *interpolation* (INT) and *out-of-input-distribution* (OOID) performance of our RCNP models with different CNP baselines on mixture function with varying input dimensions. We use normalized log-likelihoods as our metric and show mean and (standard deviation) obtained from 10 runs with different seeds. *Trivial* refers to a model that predicts a Gaussian distribution utilizing the empirical mean and standard deviation derived from the context outputs. "F" denotes failed attempts that yielded very bad results. Missing entries could not be run. Statistically significantly best results are **bolded**.

| | $d_x = 1$ | | $d_x = 2$ | | $d_x = 3$ | | $d_x = 5$ | | $d_x = 10$ | |
|---|---|---|---|---|---|---|---|---|---|---|
| | INT | OOID | INT | OOID | INT | OOID | INT | OOID | INT | OOID |
| RCNP (sta) | 0.20 (0.01) | 0.20 (0.01) | 0.17 (0.00) | 0.17 (0.00) | -0.10 (0.00) | -0.10 (0.00) | -0.31 (0.03) | -0.31 (0.03) | -0.32 (0.00) | -0.32 (0.00) |
| RGNP (sta) | 0.34 (0.03) | 0.34 (0.03) | 0.46 (0.02) | 0.46 (0.02) | **0.37** (0.01) | **0.37** (0.01) | 0.04 (0.02) | **0.04** (0.02) | **-0.11** (0.02) | **-0.11** (0.02) |
| ConvCNP | 0.38 (0.02) | 0.38 (0.02) | 0.63 (0.01) | 0.63 (0.01) | - | - | - | - | - | - |
| ConvGNP | **0.49** (0.15) | **0.49** (0.15) | **0.87** (0.03) | **0.87** (0.03) | - | - | - | - | - | - |
| CNP | 0.01 (0.01) | F | -0.22 (0.01) | -7.16 (3.61) | -0.57 (0.11) | -2.55 (1.15) | -0.72 (0.08) | -1.71 (0.55) | -0.88 (0.00) | -1.15 (0.06) |
| GNP | 0.17 (0.01) | F | -0.01 (0.01) | F | -0.17 (0.00) | -0.67 (0.05) | -0.32 (0.00) | -0.72 (0.03) | -0.38 (0.10) | -0.75 (0.09) |
| *Trivial* | -0.78 (0.00) | -0.78 (0.00) | -0.81 (0.00) | -0.81 (0.00) | -0.84 (0.00) | -0.84 (0.00) | -0.86 (0.00) | -0.86 (0.00) | -0.87 (0.00) | -0.87 (0.00) |

## E.2 Data

The datasets used to train the neural process models were generated based on synthetic functions sampled from a Gaussian process model with kernel $k$, with context and target set sizes as described in Section D.2. However, while the datasets used in the synthetic regression experiments were generated using fixed kernel setups, the current experiment explored sampling from a set of base kernels. To explore four different training regimes for each $d_x = \{3, 6\}$, we changed how the GP kernel $k(z) = \theta k_0(z/\ell)$ with an output scale $\theta$ and a lengthscale $\ell$ was chosen as follows.

*(i)* **Matérn-fixed.** The kernel $k$ is a Matérn-$\frac{5}{2}$ kernel with fixed $\ell = \sqrt{d_x}/4$ and $\theta = 1$.

*(ii)* **Matérn-sampled.** The kernel $k$ is a Matérn-$\frac{5}{2}$ kernel, with

$$\ell \sim \mathcal{LN}\left(\log(\sqrt{d_x}/4), 0.5^2\right), \quad \theta \sim \mathcal{LN}(0, 1),$$

where $\mathcal{LN}$ is a log-Normal distribution, i.e., $\log(\ell)$ follows a standard-Normal distribution.

*(iii)* **Kernel-single.** The kernel $k$ is sampled as

$$k_0 \sim \mathcal{U}\left(\left\{\text{EQ}, \text{Matérn-}\tfrac{1}{2}, \text{Matérn-}\tfrac{3}{2}, \text{Matérn-}\tfrac{5}{2}\right\}\right),$$
$$\ell \sim \mathcal{LN}\left(\log(\sqrt{d_x}/4), 0.5^2\right),$$
$$\theta \sim \mathcal{LN}(0, 1),$$

where $\mathcal{U}$ is a uniform distribution over the set.

*(iv)* **Kernel-multiple.** The kernel $k$ is sampled as

$$k_1, k_2, k_3, k_4 \sim \mathcal{U}\left(\left\{\text{NA}, \text{EQ}, \text{Matérn-}\tfrac{1}{2}, \text{Matérn-}\tfrac{3}{2}, \text{Matérn-}\tfrac{5}{2}\right\}\right),$$
$$\ell_i \sim \mathcal{LN}\left(\log(\sqrt{d_x}/4), 0.5^2\right), \qquad i \in \{1, \ldots 4\},$$
$$\theta_j \sim \mathcal{LN}(0, 1), \qquad j \in \{1, 2\},$$
$$k = \theta_1 k_1(\ell_1) \cdot k_2(\ell_2) + \theta_2 k_3(\ell_3) \cdot k_4(\ell_4),$$

Table S8: An ablation study conducted on RCNPs and FullRCNPs using synthetic regression data. We evaluated both the 'stationary' (sta) version, which employs the difference comparison function, and the 'isotropic' (iso) version, utilizing the distance comparison function, across both RCNP and FullRCNP models. The table provides metrics for both the *interpolation* (INT) and *out-of-input-distribution* (OOID) performance. We show mean and (standard deviation) obtained from 10 runs with different seeds. Statistically significantly best results are **bolded**.

| | | EQ KL divergence($\downarrow$) | | | Matérn-$\frac{5}{2}$ KL divergence($\downarrow$) | | |
|---|---|---|---|---|---|---|---|
| | | $d_x = 2$ | $d_x = 3$ | $d_x = 5$ | $d_x = 2$ | $d_x = 3$ | $d_x = 5$ |
| INT | RCNP (sta) | 0.26 (0.00) | 0.40 (0.01) | 0.45 (0.00) | 0.30 (0.00) | 0.39 (0.00) | 0.35 (0.00) |
| | RGNP (sta) | **0.03** (0.00) | 0.05 (0.00) | 0.11 (0.00) | **0.03** (0.00) | 0.05 (0.00) | 0.11 (0.00) |
| | RCNP (iso) | 0.32 (0.00) | 0.41 (0.00) | 0.46 (0.00) | 0.34 (0.00) | 0.39 (0.00) | 0.36 (0.00) |
| | RGNP (iso) | 0.12 (0.00) | 0.17 (0.00) | 0.30 (0.00) | 0.12 (0.00) | 0.17 (0.00) | 0.24 (0.00) |
| | FullRCNP (sta) | 0.25 (0.00) | 0.30 (0.00) | 0.36 (0.00) | 0.30 (0.00) | 0.31 (0.00) | 0.29 (0.00) |
| | FullRGNP (sta) | **0.03** (0.00) | **0.04** (0.00) | **0.09** (0.00) | **0.03** (0.00) | **0.04** (0.00) | **0.10** (0.00) |
| | FullRCNP (iso) | 0.26 (0.00) | 0.31 (0.00) | 0.35 (0.00) | 0.30 (0.00) | 0.32 (0.00) | 0.29 (0.00) |
| | FullRGNP (iso) | 0.08 (0.00) | 0.14 (0.00) | 0.25 (0.00) | 0.09 (0.00) | 0.16 (0.00) | 0.21 (0.00) |
| OOID | RCNP (sta) | 0.26 (0.00) | 0.40 (0.01) | 0.45 (0.00) | 0.30 (0.00) | 0.39 (0.00) | 0.35 (0.00) |
| | RGNP (sta) | **0.03** (0.00) | 0.05 (0.00) | 0.11 (0.00) | **0.03** (0.00) | 0.05 (0.00) | 0.11 (0.00) |
| | RCNP (iso) | 0.32 (0.00) | 0.41 (0.00) | 0.46 (0.00) | 0.34 (0.00) | 0.39 (0.00) | 0.36 (0.00) |
| | RGNP (iso) | 0.12 (0.00) | 0.17 (0.00) | 0.30 (0.00) | 0.12 (0.00) | 0.17 (0.00) | 0.24 (0.00) |
| | FullRCNP (sta) | 0.25 (0.00) | 0.30 (0.00) | 0.36 (0.00) | 0.30 (0.00) | 0.31 (0.00) | 0.29 (0.00) |
| | FullRGNP (sta) | **0.03** (0.00) | **0.04** (0.00) | **0.09** (0.00) | **0.03** (0.00) | **0.04** (0.00) | **0.10** (0.00) |
| | FullRCNP (iso) | 0.26 (0.00) | 0.31 (0.00) | 0.35 (0.00) | 0.30 (0.00) | 0.32 (0.00) | 0.29 (0.00) |
| | FullRGNP (iso) | 0.08 (0.00) | 0.14 (0.00) | 0.25 (0.00) | 0.09 (0.00) | 0.16 (0.00) | 0.21 (0.00) |

where NA indicates that no kernel is chosen, i.e., a term in the sum can consist of a single kernel or be missing completely. This setup is based on the kernel learning setup in [35].

See Section D.2 for the precise definition of each kernel. Note that each setting is strictly more general than the one before and entails its predecessor as a special case. Setting *(iv)* was reported in the main text.

### E.3  Bayesian optimization

After training, the neural process models were evaluated in a Bayesian optimization task. Starting from a common set of five random initial observations, $\mathbf{x}_0 \in \mathcal{U}(0,1)^{d_x}$, each optimization run performed fifty steps of requesting queries by maximizing the expected improvement acquisition function:

$$\text{EI}(x) = \mathbb{E}\left[\max(f(x) - f_{\text{best}}, 0)\right],$$

where $f(\cdot)$ is the negative target function (as we wanted to minimize $f$), $f_{\text{best}}$ is the current observed optimum, and the expectation is calculated based on the neural process or GP model predictions. The acquisition function was optimized via PyCMA [17], the Python implementation of *Covariance Matrix Adaptation Evolution Strategy* (CMA-ES) [16]. Each CMA-ES run was initialized from the best acquisition function value estimated over a random subset of 100 locations from the hypercube. After each step, the newly queried point and its function evaluation were used to update the surrogate model. For neural process models, this means that the point was added to the context set, whereas for the the GP model, this means that the model was also reinitialized and its hyperparameters refit via type-II maximum likelihood estimation. The neural process model parameters were kept fixed without retraining throughout the Bayesian optimization task.

The results reported in Figure 2 in the main paper were computed using a single pretrained neural process model and ten random Bayesian optimization restarts for each model. For the full results presented here, each neural process model was pretrained three times and used as a surrogate model in ten optimization runs initialized with different sets of five observations, giving us a total of thirty optimization runs with each model. See Figure S1 and Figure S2 for the results and their respective discussion.

## E.4 Computation time

Each model was trained on a single GPU, requiring about $90$–$120\,\mathrm{min}$ per run. The subsequent CMA-ES optimization and analysis took about $60$–$90\,\mathrm{min}$ running on a CPU with eight cores per experiment. The total runtime was approximately $280\,\mathrm{h}$ of GPU time and approximately $7\,\mathrm{h}$ on the CPUs.

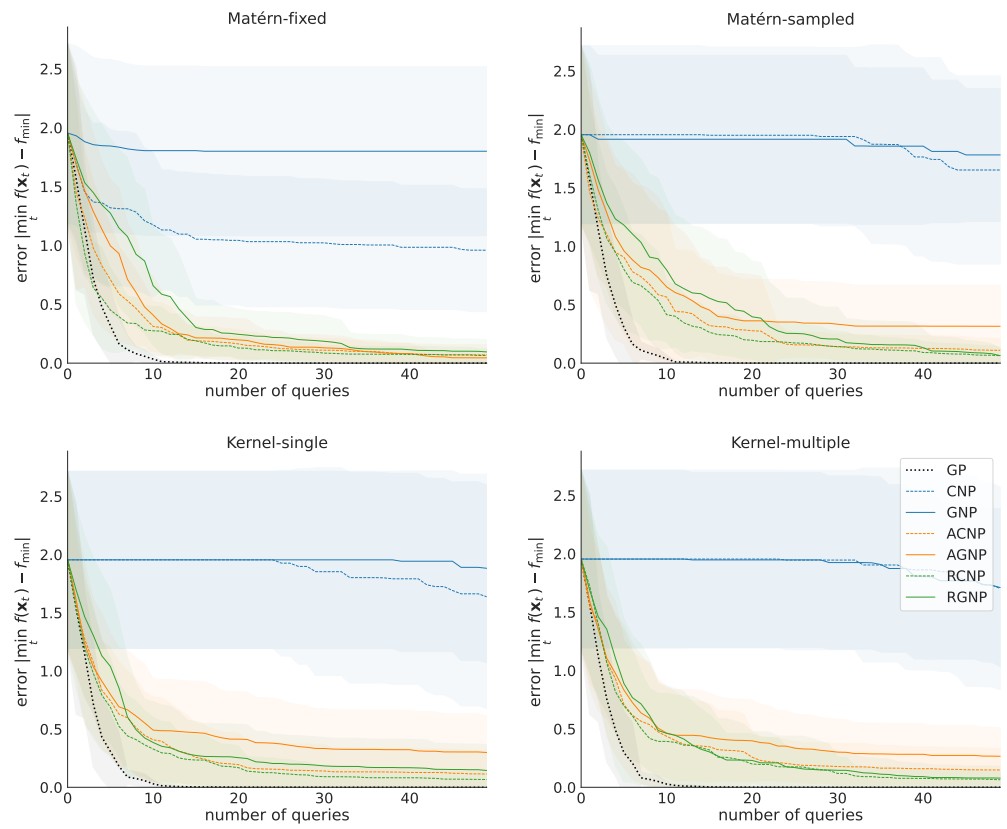

Figure S1: **Ablations on Hartmann 3d.** GNP struggled throughout all four pretraining scenarios, while its attentive and relational variants (AGNP, RGNP) were able to benefit from the additional flexibility, greatly improving their performance. CNP struggled to learn as the flexibility in the context sets increases, while its attentive and relational counterparts adapted themselves and were able to converge to optimal performance even in the most restrictive pretraining scenario.

# F  Details of Lotka–Volterra model experiments

We report here details and additional results for the Lotka–Volterra model from Section 5.3 of the main text.

## F.1  Models

The RCNP architecture used in this experiment is modeled on the multi-output CNP architecture proposed in previous work [3, Appendix F]. The multi-output CNP encodes the context set associated with each output as a 256-dimensional vector and concatenates the vectors. The encoder network used in this model is an MLP with three hidden layers and 256 hidden units per layer, and the decoder network is an MLP with six hidden layers and 512 hidden units per layer. The RCNP model replicates this architecture by encoding the targets with respect to the context set associated with each output as 256-dimensional vectors, concatenating the vectors, and using encoder and decoder networks that are the same as the encoder and decoder networks used in the CNP model. We note that the encoder

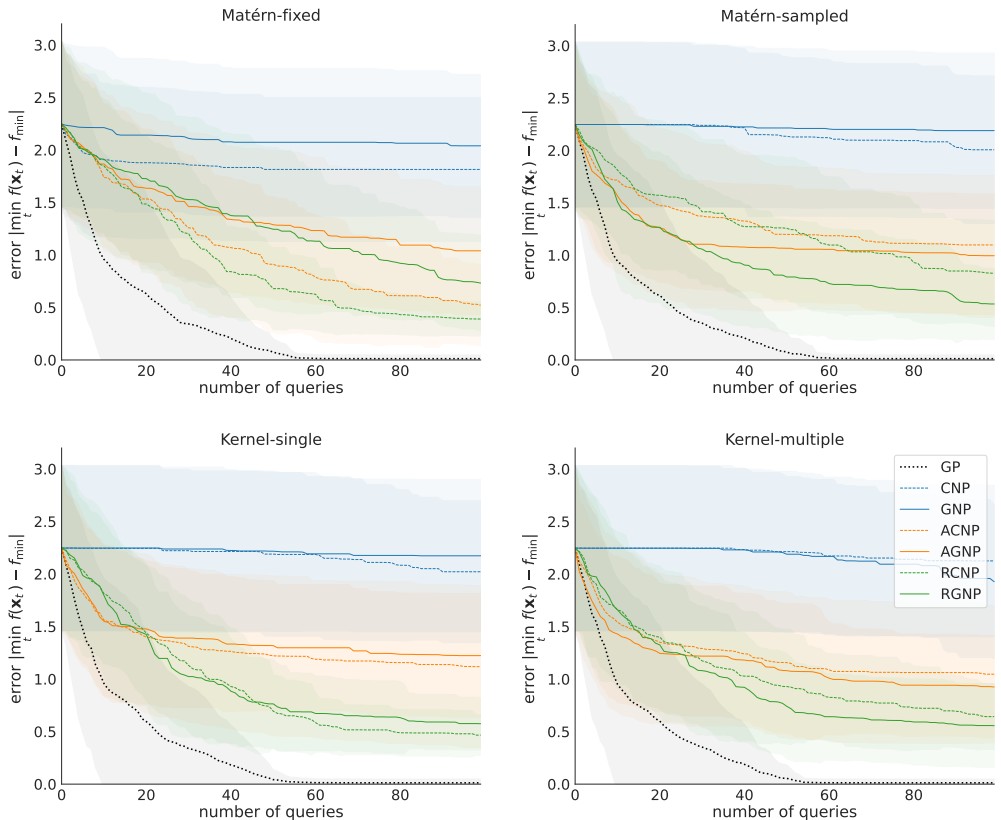

Figure S2: **Ablations on Hartmann 6d.** As in the 3d case (see Figure S1), CNP and GNP struggled to learn at all. Similarly, the advanced attentive and relational CNP models converged to their optimum even with the most restrictive prior kernel set. As the flexibility increased, RGNP was able to benefit the most and improved both upon AGNP as well as their conditional NP variants (ACNP, RCNP).

network dimensions are the same as used in the synthetic regression experiments (Section D.1) while the decoder network has more hidden units per layer in the multi-output architectures. The encoder and decoder networks use ReLU activation functions.

The models evaluated in these experiments include RCNP, CNP, ACNP, and ConvCNP as well as RGNP, GNP, AGNP, and ConvGNP. RCNP and RGNP models share the architecture described above while the other models are evaluated with the architectures proposed in previous work [3]. However, in contrast to previous work, where CNP, GNP, ACNP, and AGNP models encoded the context associated with each output with a separate encoder, here we used a shared encoder in RCNP, RGNP, CNP, GNP, ACNP, and AGNP. We made this choice based on preliminary experiments carried out with synthetic multi-output data, where we observed that using a shared encoder improved both RCNP and CNP results. We note that all models are extended with a nonlinear transformation to take into account that the population counts modeled in this experiment are positive [3].

The neural process models were trained with simulated data and evaluated with simulated and real data. The models were trained for 200 epochs with $2^{14}$ datasets in each epoch and learning rate $10^{-4}$. The validation sets used in training included $2^{12}$ datasets also generated based on the simulated data. The evaluation sets generated based on simulated data included $2^{12}$ datasets each and the evaluation sets generated based on real data $2^8$ dataset each, and both simulated and real data were used to generate three separate evaluation sets to represent the three task types considered in this experiment: interpolation, forecasting, and reconstruction.

## F.2  Data

The datasets used in this task included simulated and real datasets. The simulated datasets were constructed based on simulated predator and prey time series generated using stochastic Lotka–Volterra equations. The Lotka–Volterra equations introduced independently by Lotka [28] and Volterra [42] model the joint evolution between prey and predator populations as

$$\begin{cases} \dot{u} = \alpha u - \beta uv \\ \dot{v} = \delta uv - \gamma v \end{cases},$$

where $u$ denotes the prey, $v$ denotes the predators, $\alpha$ is the birth rate of the prey, $\beta$ is the rate at which predators kill prey, $\delta$ the rate at which predators reproduce when killing prey, and $\gamma$ is the death rate of the predators; the dot denotes a temporal derivative, i.e. $\dot{u} \equiv \frac{\mathrm{d}}{\mathrm{d}t}u$. In this work, we generated simulated data with a stochastic version proposed by [3],

$$\begin{cases} \mathrm{d}u = \alpha u\,\mathrm{d}t - \beta uv\,\mathrm{d}t + u^{1/6}\sigma\,\mathrm{d}W^u \\ \mathrm{d}v = \delta uv\,\mathrm{d}t - \gamma v\,\mathrm{d}t + v^{1/6}\sigma\,\mathrm{d}W^v \end{cases},$$

where $W^u$ and $W^v$ are two independent Brownian motions that add noise to the population trajectories and $\sigma > 0$ controls the noise magnitude.

We simulated trajectories from the above equation using the code released with the previous work [3]. The parameters used in each simulation were sampled from: $\alpha \sim \mathcal{U}(0.2, 0.8)$, $\beta \sim \mathcal{U}(0.04, 0.08)$, $\delta \sim \mathcal{U}(0.8, 1.2)$, $\gamma \sim \mathcal{U}(0.04, 0.08)$, and $\sigma \sim \mathcal{U}(0.5, 10)$. The initial population sizes were sampled from $u, v \sim \mathcal{U}(5, 100)$, and the simulated trajectories were also scaled by a factor $r \sim \mathcal{U}(1, 5)$. To construct a dataset, 150–250 observations were sampled from the simulated prey time series and 150–250 observations from the simulated predator series. The selected observations were then divided into context and target observations based on the selected task type which was fixed in the evaluation sets and selected at random in the training and validation data.

The datasets representing interpolation, forecasting, or reconstruction tasks were created based on the selected simulated observations as follows. To create an interpolation task, 100 predator and 100 prey observations, selected at random, were included in the target set, and the rest were included in the context set. To create a forecasting task, a time point was selected at random between 25 and 75 years, and all observations prior to the selected time point were included in the context set and the rest were included in the target set. To create a reconstruction task, a random choice was made between prey and predators such that the observations from the selected population were all included in the context set while the observations from the other population were divided between the context and target sets by choosing at random a time point between 25 and 75 years and including all observations prior to the selected time point in the context set and the rest in the target set.

The real datasets used to evaluate the neural process models in the sim-to-real tasks were generated based on the famous hare–lynx data [32]. The hare and lynx time series include 100 observations at regular intervals. The observations were divided into context and target sets based on selected task types, and the same observations were used to generate separate evaluation sets for each task. The forecasting and reconstruction task was created with the same approach that was used with simulated data, while the interpolation task was created by choosing at random 1–20 predator observations and 1–20 prey observations that were included in the target set and including the rest in the context set.

## F.3  Full results

We evaluate and compare the neural process models based on normalized log-likelihood scores calculated based on each evaluation set and across training runs. We compare RCNP, CNP, ACNP, and ConvCNP models in Table S9 (a) and RGNP, GNP, AGNP, and ConvGNP models with 32 basis functions in Table S9 (b) and with 64 basis functions in Table S9 (c). We observe that the best model depends on the task, and comparison across the tasks indicates that RCNP or RGNP are not consistently better or worse than their convolutional and attentive counterparts. This is not the case with CNP and GNP, and we observe that the other models including RCNP and RGNP are almost always substantially better than CNP or GNP. The models with 64 basis functions performed on average better than the models with 32 basis functions, but we observed that some RGNP, GNP, and AGNP training runs did not converge and most GNP runs did not complete 200 epochs due to numerical errors when 64 basis functions were used. We therefore reported results with 32 basis functions in the main text.

Table S9: Normalized log-likelihood scores in the Lotka–Volterra experiments (higher is better). We compared models that predict (a) mean and variance and models that also predict a low-rank covariance matrix with (b) 32 basis functions or (c) 64 basis functions. The mean and (standard deviation) reported for each model are calculated based on 10 training outcomes evaluated with the same simulated (S) and real (R) learning tasks. The tasks include *interpolation* (INT), *forecasting* (FOR), and *reconstruction* (REC). Statistically significantly best results are **bolded**. RCNP and RGNP models perform on par with their convolutional and attentive counterparts, and nearly always substantially better than the standard CNPs and GNPs, respectively.

|     |         | INT (S) | FOR (S) | REC (S) | INT (R) | FOR (R) | REC (R) |
|-----|---------|---------|---------|---------|---------|---------|---------|
| (a) | RCNP    | -3.57 (0.02) | -4.85 (0.00) | -4.20 (0.01) | -4.24 (0.02) | -4.83 (0.03) | -4.55 (0.05) |
|     | ConvCNP | **-3.47** (0.01) | **-4.85** (0.00) | **-4.06** (0.00) | **-4.21** (0.04) | -5.01 (0.02) | -4.75 (0.05) |
|     | ACNP    | -4.04 (0.06) | -4.87 (0.01) | -4.36 (0.03) | **-4.18** (0.05) | **-4.79** (0.03) | **-4.48** (0.02) |
|     | CNP     | -4.78 (0.00) | -4.88 (0.00) | -4.86 (0.00) | -4.74 (0.01) | **-4.81** (0.02) | -4.70 (0.01) |
| (b) | RGNP    | -3.51 (0.01) | **-4.27** (0.00) | -3.76 (0.00) | -4.31 (0.06) | **-4.47** (0.03) | -4.39 (0.11) |
|     | ConvGNP | **-3.46** (0.00) | -4.30 (0.00) | **-3.67** (0.01) | **-4.19** (0.02) | -4.61 (0.03) | -4.62 (0.11) |
|     | AGNP    | -4.12 (0.17) | -4.35 (0.09) | -4.05 (0.19) | -4.33 (0.15) | **-4.48** (0.06) | **-4.29** (0.10) |
|     | GNP     | -4.62 (0.03) | -4.38 (0.04) | -4.36 (0.07) | -4.79 (0.03) | -4.72 (0.04) | -4.72 (0.04) |
| (c) | RGNP    | -3.54 (0.06) | **-4.15** (0.10) | -3.75 (0.10) | -4.30 (0.04) | **-4.44** (0.02) | **-4.38** (0.10) |
|     | ConvGNP | **-3.46** (0.00) | **-4.15** (0.01) | **-3.66** (0.01) | **-4.20** (0.03) | -4.57 (0.05) | -4.64 (0.12) |
|     | AGNP    | -4.06 (0.10) | **-4.16** (0.04) | -3.95 (0.11) | -4.34 (0.10) | **-4.41** (0.06) | **-4.30** (0.08) |
|     | GNP     | -4.63 (0.03) | -4.36 (0.05) | -4.40 (0.05) | -4.82 (0.01) | -4.75 (0.04) | -4.75 (0.03) |

## F.4   Computation time

When the models were trained on a single GPU, each training run with RCNP and RGNP took around 6.5–7 hours and each training run with the reference models around 2–3 hours depending on the model. The results presented in this supplement and the main text required 120 training runs, and 10–15 training runs had been carried out earlier to confirm as a sanity check our ability to reproduce the results presented in previous work. The multi-output models used in this experiment were additionally studied in preliminary experiments using synthetic regression data. The computation time used in all the experiments is included in the estimated total reported in Section C.2.

# G   Details of Reaction-Diffusion model experiments

We report here details and additional results for the Reaction-Diffusion model from Section 5.4 of the main text.

## G.1   Models

We compared RCNP, RGNP, ACNP, AGNP, CNP, and GNP models trained using simulated data with input dimensions $d_x = \{3, 4\}$. We mostly used the architectures described in Section D.1, but since less training data was used in this experiment, we reduced the number of hidden layers in the decoder network to 4. The models were trained for 100 epochs with $2^{10}$ datasets in each epoch and learning rate $10^{-4}$. The validation sets used in training included $2^8$ datasets and the evaluation sets used to compare the models in completion and forecasting tasks included $2^8$ datasets each.

## G.2   Data

We generated simulated data using a simplified equation inspired by the cancer evolution equation from Gatenby and Gawlinski [13]. The equation involves three quantities: the healthy cells $(\mathbf{z}, t) \mapsto u(\mathbf{z}, t)$, the cancerous cells $(\mathbf{z}, t) \mapsto v(\mathbf{z}, t)$, and an acidity measure $(\mathbf{z}, t) \mapsto w(\mathbf{z}, t)$, with

$t \in [0, T]$ and $\mathbf{z} \in E \subset \mathbb{R}^2$. Their temporal and spatial dynamics are described by:

$$\begin{cases} \dot{u} = r_u u(1 - u/k_u) - d_u w \\ \dot{v} = r_v v(1 - v/k_v) + D_v \nabla^2 v \\ \dot{w} = r_w v - d_w w + D_w \nabla^2 v \end{cases} , \tag{S8}$$

where $r_u$, $r_v$, and $r_w$ are apparition rates, $k_u$ and $k_v$ control the maximum for the number of healthy and cancerous cells, $d_u$ and $d_w$ are death rates relative to other species, and $D_v$ and $D_w$ are dispersion rates.

We simulate trajectories from Eq. S8 on a discretized space-time grid using the SpatialPy simulator [6]. For the purpose of our experiment, we selected realistic parameter ranges that would also produce quick evolution. The selected parameter ranges are therefore close to [13], but for example the diffusion rates were increased. The parameters for the simulations are sampled according to the following distributions: $k_u = 10$, $k_v = 100$, $r_u \sim \mathcal{U}(0.0027, 0.0033)$, $r_v \sim \mathcal{U}(0.027, 0.033)$, $r_w \sim \mathcal{U}(1.8, 2.2)$, $D_v \sim U(0.0009, 0.0011)$, $D_w \sim \mathcal{U}(0.009, 0.011)$, $d_w = 0$, $d_u \sim \mathcal{U}(0.45, 0.55)$. These parameters lead to a relatively slowly-growing number of cancerous cells, with a fast decay in healthy cells due to acid spread, as depicted in Figure S3. The process starts with the healthy cells count $u$ constant across the whole space, cancerous cells count $v$ zero except for one cancerous cell at a uniformly random position, and acid count $w$ zero across the whole space.

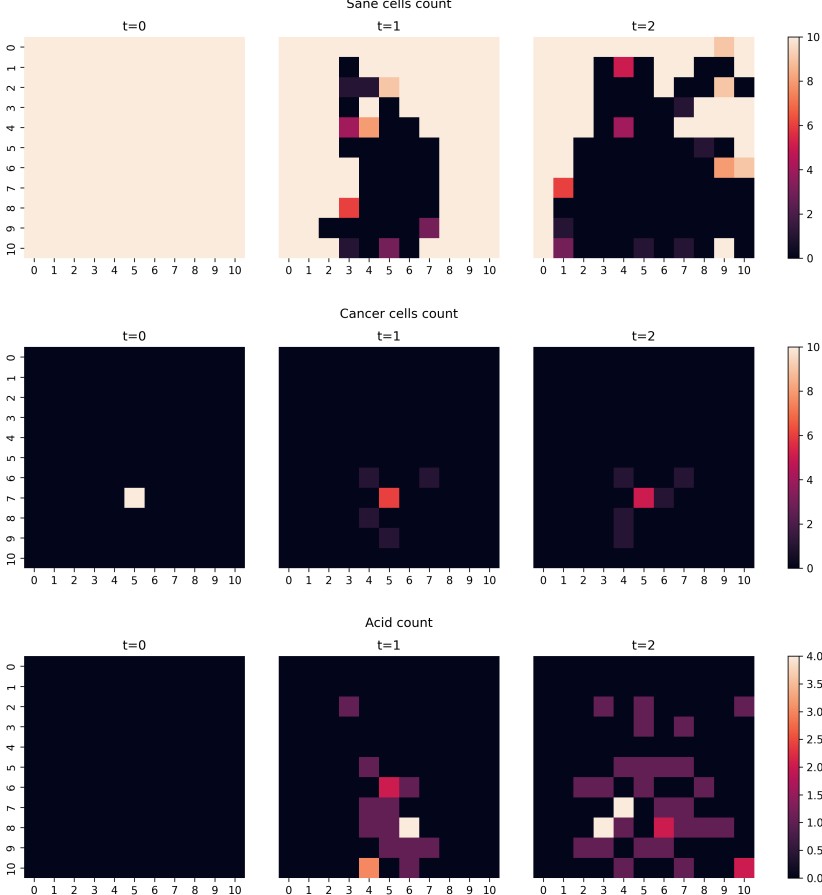

Figure S3: **Example simulation of the Reaction-Diffusion model at times** $t = 0, 1, 2$. Each row represent a different quantity in the model (from top to bottom, healthy cells, cancerous cells, and acid). At time $t = 0$, the population of healthy cells is 10 at each point of space, and a single point is occupied by additional cancerous cells. After this time, cancerous cells start spreading and slowly multiplicating, while acid spreads faster, killing healthy cells along its way.

We generated a total $5 \cdot 10^3$ trajectories on a $11 \times 11$ grid at times $t = 0, \ldots, 5$. We used the first $4 \cdot 10^3$ trajectories to generate datasets for model training and the other $10^3$ to generate datasets for evaluation. The datasets were generated by sampling from the trajectories as follows. First, we extracted the population counts at three consecutive time points from a trajectory and time point selected at random. We then sampled 50–200 context points and 100–200 target points from this subsample to create a dataset representing either a completion or forecasting task. For the completion task, the context and target points were sampled from the subsample at random, and for the forecasting task, all context points were sampled from the first two time points, and all target points from the third time point were included in the subsample. All datasets included in the training and validation data represented the completion task.

Finally, the input and output variables at each context and target point were determined as follows. In the experiments reported in the main text, we considered the case of partial observations, where only the difference $u - v$ is observed. This makes the simulator model a hidden Markov model and thus intractable. Predicting future states of the quantity $u - v$ without running inference and without having access to $w$ is a challenging problem, similar to several application cases of interest. In this appendix, we additionally consider a scenario where we include the acid count in the input variables and take the healthy cells count as the output variable.

### G.3 Additional results

A promising scientific application of neural processes is inferring latent variables given other measured quantities. To explore this capability, we attempt to infer the quantity of healthy cells ($u(\mathbf{z}, t)$) given as input the acidity concentration ($w(\mathbf{z}, t)$) for some $\mathbf{z}, t$. For this study, we augmented the inputs to $(\mathbf{z}, t, w(\mathbf{z}, t))$. Notably, the regression problem is not equivariant in the last input dimension, so we encoded $w(\mathbf{z}, t)$ in a non-relational manner also in the RCNP and RGNP models. The results are presented in Table S10. The difference between the completion and forecasting tasks is small in this experiment, and the best results are observed with RGNP in both tasks. AGNP performance is surprisingly poor and will require further investigation.

Table S10: Normalized log-likelihood scores in the additional experiments of the Reaction-Diffusion problem for both tasks (higher is better). Mean and (standard deviation) from 10 training runs evaluated on a separate test dataset.

|  | RCNP | RGNP | ACNP | AGNP | CNP | GNP |
|---|---|---|---|---|---|---|
| Completion | 0.35 (0.12) | **0.74** (0.18) | 0.24 (0.25) | -2.20 (3.67) | 0.44 (0.17) | 0.32 (0.26) |
| Forecasting | 0.29 (0.11) | **0.63** (0.15) | 0.38 (0.21) | -1.7 (3.0) | 0.46 (0.13) | 0.41 (0.20) |

### G.4 Computation time

On GPU, for all methods the training runs of this experiment were particularly fast, each one lasting no more than 5 minutes. We carried out a total of 10 training runs for each method, with a total of 120 training runs. We also performed around 60 exploratory training runs (with longer training time, larger networks, *etc.*). The total training time comes to less than 15 hours.

## H   Additional experiments

In addition to the experimental evaluation in the main paper and their extended discussion in the sections above, we explored several further experimental setups to demonstrate the wide applicability of our proposal. These are *(i)* an application of the autoregressive structure proposed by [3] to our proposed approach (Section H.1), *(ii)* a demonstration of how to incorporate rotational equivariance into the model (Section H.2), and *(iii)* an exploration of the method's performance on two popular image data regression tasks (Section H.3).

### H.1   Autoregressive CNPs

To further demonstrate the effectiveness of our RCNP architectures, we ran a set of experiments adopting an autoregressive approach to both our RCNP and other CNP models. Autoregressive

CNPs (AR-CNPs) are a new family of CNP models which were recently proposed by [3]. Unlike traditional CNPs, which generate predictions independently for each target point, or GNPs, which produce a joint multivariate normal distribution over the target set, AR-CNPs define a joint predictive distribution autoregressively, leveraging the chain rule of probability. Without changing the CNP architectures and the training procedure, AR-CNPs feed earlier output predictions back into the model autoregressively to predict new points. This adjustment enables the AR-CNPs to capture highly dependent, non-Gaussian predictive distributions, resulting in better performance compared to traditional CNPs.

In this experiment, we compare AR-RCNP, AR-CNP, and AR-ConvCNP on synthetic regression data. The AR approach is applied at the evaluation stage to generate the target predictions autoregressively. When applying AR, the training method for the models remains unchanged. Thus, for this experiment, we use the same models that were trained with synthetic data. Details about the data generation and model configurations can be found in Section D. We chose the translational-equivariant version of RCNP for each task.

The full results are reported in Table S11. As we maintain consistent experimental settings with previous synthetic experiments (Section D), we can directly compare the performance of models without the AR approach (Table S3-S7) to those enhanced with the AR approach. We observe that the AR approach improves RCNP performance in the Gaussian experiments in all dimensions and in the non-Gaussian sawtooth and mixture experiments when $d_x > 1$. When $d_x = 1$, AR-RCNP results are comparable to RCNP results in the mixture task and worse in the sawtooth task. We also observe that AR-CNP results are worse than CNP results in most sawtooth and mixture experiments. Overall we can see that AR-RCNPs are better than AR-CNPs across all dimensions and tasks under both INT and OOID conditions, and achieve close performance compared with AR-ConvCNPs under $d_x = 1$ in the Gaussian experiments.

Table S11: Comparison of the *interpolation* (INT) and *out-of-input-distribution* (OOID) performance of AR-RCNP model with other AR baselines on various synthetic regression tasks with varying input dimensions. We show the mean and (standard deviation) obtained from 10 runs with different seeds. "F" denotes failed attempts that yielded very bad results. Our AR-RCNP method performs competitively in low dimension, and scales to higher input dimensions ($d_x > 2$), where AR-ConvCNP is not applicable.

| | | EQ KL divergence(↓) | | Matérn-$\frac{5}{2}$ KL divergence(↓) | | Weakly-periodic KL divergence(↓) | | Sawtooth log-likelihood(↑) | | Mixture log-likelihood(↑) | |
|---|---|---|---|---|---|---|---|---|---|---|---|
| | | INT | OOID | INT | OOID | INT | OOID | INT | OOID | INT | OOID |
| $d_x = 1$ | AR-RCNP | 0.02 (0.00) | 0.02 (0.00) | 0.02 (0.00) | 0.02 (0.00) | 0.04 (0.00) | 0.04 (0.00) | 1.93 (0.05) | 1.97 (0.04) | 0.20 (0.27) | 0.20 (0.27) |
| | AR-ConvCNP | **0.01** (0.00) | **0.01** (0.00) | **0.00** (0.00) | **0.00** (0.00) | **0.01** (0.00) | **0.01** (0.00) | **4.11** (0.09) | **4.11** (0.09) | **0.83** (0.03) | **0.82** (0.03) |
| | AR-CNP | 0.09 (0.01) | 4.33 (3.44) | 0.12 (0.01) | F | 0.21 (0.01) | 5.05 (4.52) | -1.77 (1.20) | F | -0.25 (0.09) | F |
| $d_x = 3$ | AR-RCNP | **0.14** (0.01) | **0.14** (0.01) | **0.13** (0.00) | **0.13** (0.00) | **0.11** (0.00) | **0.11** (0.00) | **1.55** (0.01) | **1.55** (0.01) | **0.20** (0.00) | **0.20** (0.00) |
| | AR-CNP | 0.29 (0.00) | 3.05 (1.26) | 0.32 (0.03) | 2.24 (1.20) | 0.33 (0.00) | 2.03 (0.75) | 0.25 (0.33) | -0.26 (0.09) | -0.39 (0.01) | -3.72 (1.68) |
| $d_x = 5$ | AR-RCNP | **0.23** (0.00) | **0.23** (0.00) | **0.15** (0.00) | **0.15** (0.00) | **0.16** (0.00) | **0.16** (0.00) | **0.87** (0.01) | **0.87** (0.01) | **-0.10** (0.00) | **-0.10** (0.00) |
| | AR-CNP | 0.49 (0.00) | 1.22 (0.05) | 0.39 (0.00) | 0.97 (0.05) | 0.38 (0.00) | 1.47 (0.35) | -0.00 (0.13) | F | -0.66 (0.08) | -3.51 (2.32) |

### H.1.1 Computation time

The AR procedure is not used in model training, so the training procedure aligns with that of standard CNP models. The experiments reported in this section used AR with models trained for the synthetic regression experiments in Section 5.1. If the models had been trained from scratch, we estimate that the total computational cost for these experiments would have been approximately 2000 GPU-hours. AR is applied in the evaluation phase, where we predict target points autoregressively. This process is considerably lengthier than standard evaluation, with the duration influenced by both the target set size and data dimension. Nonetheless, since only a single pass is required during evaluation, the overall computational time remains comparable to standard CNP models.

### H.2 Rotation equivariance

The experiments presented in this section use a two-dimensional toy example to investigate modeling datasets that are rotation equivariant but not translation equivariant. We created regression tasks based

on synthetic data sampled from a GP model with an exponentiated quadratic covariance function and mean function $m(\mathbf{x}) = \|\mathbf{ARx}\|_2^2$, where $\mathbf{A} = \text{diag}(\mathbf{a})$ is a fixed diagonal matrix with unequal entries and $\mathbf{R}$ is a random rotation matrix. We considered an isotropic-kernel model version with the standard EQ covariance function (Equation S3) and an anisotropic-kernel model version with the covariance function $k(\mathbf{x}, \mathbf{x}') = \exp(-\|\mathbf{BRx} - \mathbf{BRx}'\|_2^2)$, where $\mathbf{B} = \text{diag}(\mathbf{b})$ is a fixed diagonal matrix with unequal entries and $\mathbf{R}$ is the same random rotation matrix as in the mean function. Both models use the anisotropic GP mean function defined previously.

We generated the datasets representing regression tasks by sampling context and target points from the synthetic data generated with isotropic-kernel or anisotropic-kernel model version as follows. The number of context points sampled varied uniformly between 1 and 70, while the number of target points was fixed at 70. All training and validation datasets were sampled from the range $\mathcal{X} = [-4, 0] \times [-4, 0]$. To evaluate the models in an interpolation (INT) task, we generated evaluation datasets by sampling context and target points from the training range. To evaluate the models in an out-of-input distribution (OOID) task, we generated evaluation datasets with context and target points sampled in the range $[0, 4] \times [0, 4]$.

We chose a comparison function $g$ that is rotation invariant but not translation invariant defined as[2]

$$g(\mathbf{x}, \mathbf{x}') = (\|\mathbf{x} - \mathbf{x}'\|_2, \|\mathbf{x}\|_2, \|\mathbf{x}'\|_2).$$

Since 'simple' RCNPs are only context preserving for translation-equivariance and not for other equivariances (Proposition 4.9), we combine the rotation-equivariant comparison function with a FullRCNP model.

We compared FullRCNP with the rotation-equivariant comparison function, FullRCNP (rot), with CNP and ConvCNP models in both the isotropic-kernel and anisotropic-kernel test condition. We also used isotropic-kernel data to train and evaluate a translation-equivariant ('stationary') RCNP version, RCNP (sta), that uses the difference comparison function. The model architectures and training details were the same as in the two-dimensional synthetic regression experiments (Section D.1).

The results reported in Table S12 show that, as expected, the isotropic-kernel test condition is easier than the anisotropic-kernel test condition where the random rotations change the covariance structure. The best results in both conditions are observed with ConvCNP when the models are evaluated in the interpolation task and with FullRCNP (rot) when the models are evaluated in the OOID task. Since the input range in the OOID task is rotated compared to the training range, FullRCNP (rot) results are on the same level in both tasks while the other models are not able to make reasonable predictions in the OOID task. We also observe that that the FullRCNP (rot) results in the interpolation task are better than CNP or RCNP (sta) results in the isotropic-kernel test condition and similar to CNP in the anisotropic-kernel test condition.

Table S12: Normalized log-likelihood scores in the experiments using synthetic data generated with isotropic and anisotropic GP kernels and an anisotropic GP mean function. The mean and (standard deviation) reported for each model are calculated based on 10 training outcomes evaluated with separate datasets representing *interpolation* (INT) and *out-of-input-distribution* (OOID) tasks. "F" denotes failed attempts with log-likelihood scores below $-60$.

|  | Isotropic GP kernel | | Anisotropic GP kernel | |
|---|---|---|---|---|
|  | INT | OOID | INT | OOID |
| FullRCNP (rot) | -0.89 (0.04) | **-0.90** (0.04) | -6.37 (0.34) | **-6.38** (0.31) |
| RCNP (sta) | -1.16 (0.03) | F | — | — |
| ConvCNP | **-0.60** (0.02) | F | **-2.83** (0.12) | F |
| CNP | -1.42 (0.03) | F | -6.43 (1.13) | F |

## H.2.1 Computation time

The models were trained on a single GPU, and each training run took around 6 hours with FullRCNP, 2 hours with CNP, 4 hours with ConvCNP and 3 hours with RCNP. The results provided in this

---

[2]This comparison function is invariant to proper and improper rotations around the origin (i.e., including mirroring).

appendix required 70 training runs and consumed around 250 GPU hours. Additional experiments related to this example consumed around 2500 GPU hours.

### H.3 Image datasets

The experiments reported in the main paper demonstrate how RCNP can leverage prior information about equivariances in regression tasks including tasks with more than 1–2 input dimensions. In this section, we provide details and results from additional experiments that used MNIST [24] and CelebA [27] image data to create regression tasks. The neural process models were used in these experiments to predict the mean and variance over pixel values across the two-dimensional image given some pixels as observed context. The experiments investigate how RCNP compares to other conditional neural process models when the assumption about task equivariance is incorrect.

#### H.3.1 Setup

The experiments reported in this section compared RCNP, CNP, and ConvCNP models using interpolation tasks generated based on $16 \times 16$ pixel images. The pixels were viewed as data points with $d_x = 2$ input variables that indicate the pixel location on image canvas and $d_y$ output variables that encode the pixel value. The output dimension depended on the image type. Pixels in grayscale images were viewed as data points with one output variable ($d_y = 1$) that takes values between 0 and 1 while pixels in RGB images were viewed as data points with three output variables ($d_y = 3$) that each take values between 0 and 1 to encode one RGB channel. We evaluated the selected models using standard MNIST and CelebA images (Section H.3.2) and examined modeling translation equivariance with toroidal MNIST images (Section H.3.3).

The model architectures used in the image experiments match the architectures used in the synthetic regression experiments (Section D.1) extended with an output transformation to bound the predicted mean values between 0 and 1. While we used images with multiple output channels, there was no need to use a multi-output architecture such as discussed in Section F.1. This is because we assumed that all output channels are observed when a pixel is included in the context set and unobserved otherwise, meaning that the output channels in image data are not associated with separate context sets. RCNP models were used with the standard stationary comparison function unless otherwise mentioned.

The models were trained for 200 epochs with $2^{14}$ datasets in each epoch and learning rate $3 \cdot 10^{-4}$, and tested with validation and evaluation sets that included $2^{12}$ datasets each. The log-likelihood score that is used as training objective is usually calculated based on the predicted mean and variance, but when the models were trained on MNIST data, we started training with the predicted variance fixed to a small value. This was done to prevent the neural process models from learning to predict the constant background values with low variance and ignoring the image details. We trained with the fixed variance between epochs 1–150 and with the predicted variance as usual between epochs 151–200.

The datasets used to train and evaluate the neural process models were generated as follows. Each dataset was generated based on an image that was sampled from the selected image set and downscaled to $16 \times 16$ pixels. The downscaled images were either used as such or transformed with random translations as explained in Section H.3.3. The context sets generated based on the image data included 2–128 pixels sampled at random while the target sets included all 256 pixels. The evaluation sets used in each experiment were generated with the context set size fixed to 2, 20, 40, and 100 pixels.

#### H.3.2 Experiment 1: Centered images (no translation)

The experiments reported in this section used MNIST and CelebA images scaled to the size $16 \times 16$ (Figure S4). The normalized log-likelihood scores calculated based on each evaluation set and across training runs for each neural process model are reported in Table S13. The best results are observed with ConvCNP, while the order between RCNP and CNP depends on the context set size. CNP results are better than RCNP results when predictions are generated based on $N = 2$ context points while RCNP results are better when more context data is available.

We believe CNP works better than RCNP when $N = 2$ because the absolute pixel locations are generally informative about the possible pixel values in the MNIST and CelebA images. This means

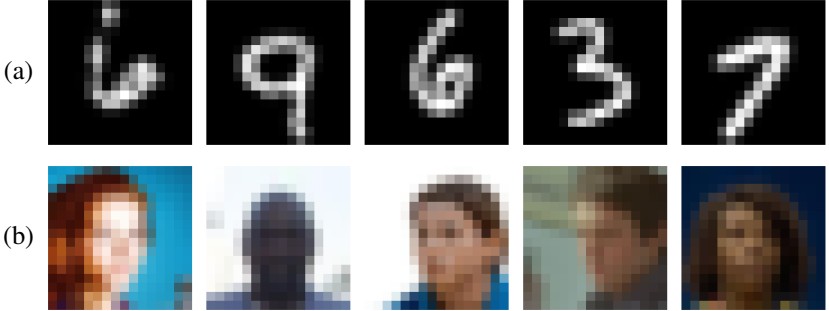

Figure S4: Example (a) MNIST and (b) CelebA $16 \times 16$ images.

Table S13: Normalized log-likelihood scores in the image experiments using (a) $16 \times 16$ MNIST images or (b) $16 \times 16$ CelebA images (higher is better). The mean and (standard deviation) reported for each model are calculated based on 10 training outcomes evaluated in interpolation task with context sizes 2, 10, 20, and 100. RCNP results are the lowest when the context set is small but improve when the context size increases.

|     |         | 2 | 20 | 40 | 100 |
|-----|---------|------------|------------|------------|------------|
| (a) | RCNP    | 5.52 (0.30) | 7.13 (0.10) | 8.08 (0.13) | 9.35 (0.15) |
|     | ConvCNP | **6.43** (0.02) | **7.35** (0.04) | **8.27** (0.04) | **9.87** (0.06) |
|     | CNP     | 6.29 (0.12) | 7.04 (0.03) | 7.47 (0.03) | 8.00 (0.05) |
| (b) | RCNP    | 0.18 (0.00) | 1.19 (0.01) | 1.79 (0.01) | 3.01 (0.03) |
|     | ConvCNP | **0.33** (0.00) | **1.36** (0.01) | **2.03** (0.01) | **3.37** (0.03) |
|     | CNP     | 0.31 (0.00) | 0.90 (0.01) | 1.09 (0.01) | 1.25 (0.01) |

that the regression tasks generated based on these images are not translation equivariant and the assumption encoded in the RCNP comparison function is incorrect. Since RCNP does not encode the absolute target location, it needs context data to make predictions that CNP and ConvCNP can make based on the target location alone (Figure S5).

While RCNP needs context data to work around the incorrect assumption about translation equivariance, RCNP results are better than CNP results when more context data is available. We believe this is because the context set representation in CNP does not depend on the current target location while RCNP can learn to preserve context information that is relevant to the current target. For example, while CNP looses information about the exact observed values when the context set is encoded, RCNP generally learns to reproduce the observed context. RCNP may also learn to emphasize context points that are close to the current target and may learn local features that are translation equivariant.

### H.3.3 Experiment 2: Translated images

The interpolation tasks in the previous image experiments do not exhibit translation equivariance since the input range is fixed and the image content is centered on the canvas. To run an image experiment with translation equivariance, we converted the $16 \times 16$ MNIST images into toruses and applied random translations to move image content around. The translations do not change the relative differences between image pixels on the torus, but when the torus is viewed as an image, both the absolute pixel locations and relative differences between pixel locations are transformed. This means that to capture the translation equivariance, we needed to calculate differences between pixel locations on the torus.

The experiments reported in this section compare models trained using either centered image data (Figure S4 (a)) or image data with random translations on the torus (Figure S6). We ran experiments with both RCNP (sta) that calculates the difference between pixel locations on the image canvas and RCNP (tor) that calculates the difference between pixel locations on the torus. The models trained with centered image data were evaluated with both centered and translated data while the models

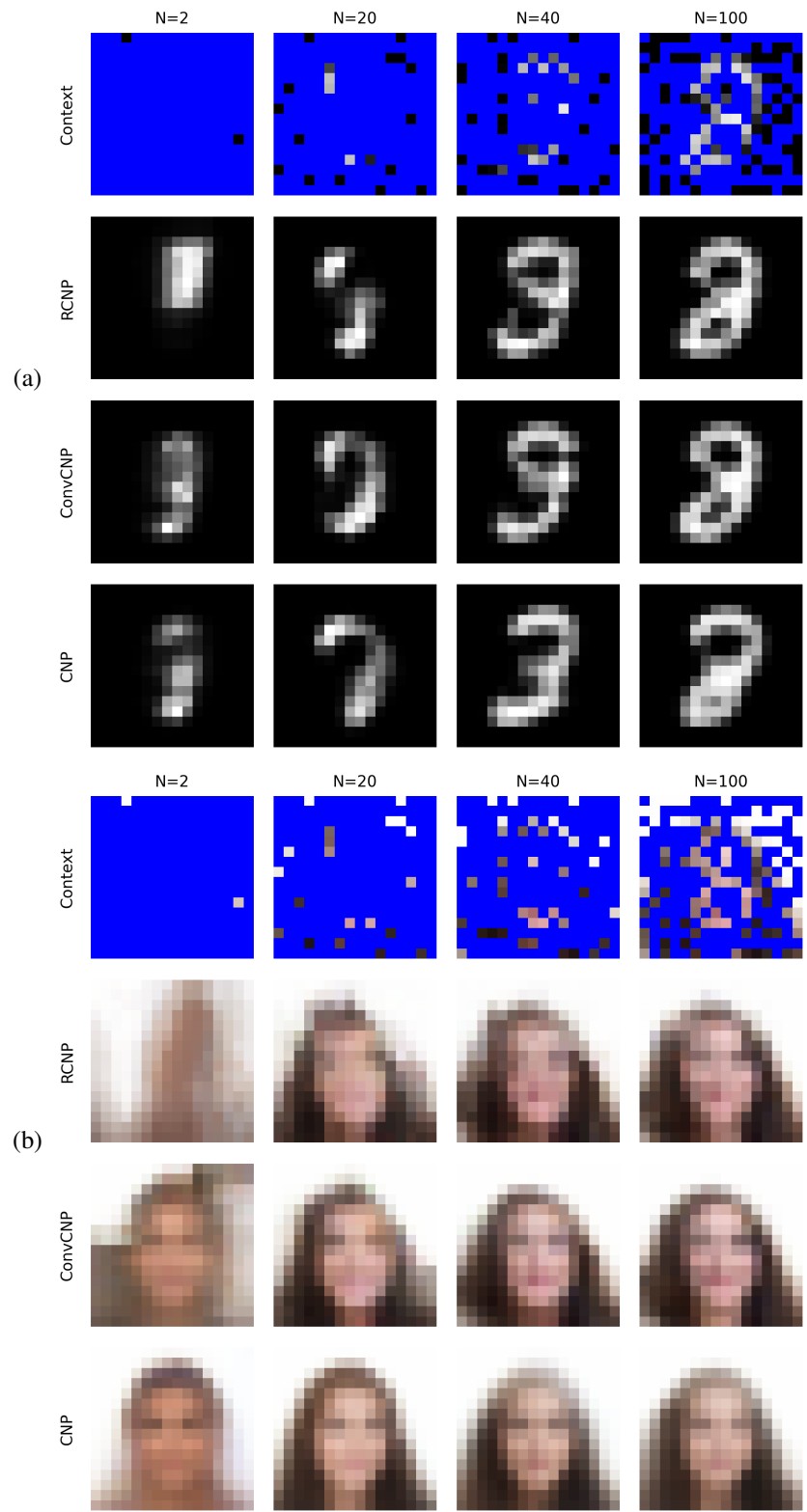

Figure S5: Example context sets and predicted mean values in the image experiment using (a) MNIST and (b) Celeba $16 \times 16$ data.

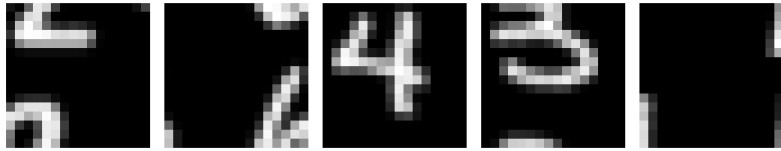

Figure S6: MNIST $16 \times 16$ example images with random translations on a torus.

trained with translated data were evaluated with translated data. The normalized log-likelihood scores are reported in Table S14.

Table S14: Normalized log-likelihood scores in the image experiments using $16 \times 16$ MNIST images with random translations on a torus (higher is better). We compare (a) models trained on centered image data evaluated on centered image data, (b) models trained on centered image data evaluated on image data with random translations, and (c) models trained on image data with random translations evaluated on image data with random translations. The mean and (standard deviation) reported for each model are calculated based on 10 training outcomes evaluated in interpolation task with context sizes 2, 10, 20, and 100. "F" denotes very bad results with negative log-likelihood scores. RCNP (tor) trained on centered image data is able to make predictions about images with random translations because the translation mechanism is modeled in the comparison function.

|     |           | 2           | 20          | 40          | 100         |
| --- | --------- | ----------- | ----------- | ----------- | ----------- |
|     | RCNP (sta) | 5.52 (0.30) | 7.13 (0.10) | 8.08 (0.13) | 9.35 (0.15) |
| (a) | RCNP (tor) | 5.51 (0.01) | 6.63 (0.04) | 7.59 (0.03) | 9.15 (0.09) |
|     | ConvCNP   | **6.43** (0.02) | **7.35** (0.04) | **8.27** (0.04) | **9.87** (0.06) |
|     | CNP       | 6.29 (0.12) | 7.04 (0.03) | 7.47 (0.03) | 8.00 (0.05) |
|     | RCNP (sta) | F           | F           | F           | F           |
| (b) | RCNP (tor) | **5.62** (0.01) | **6.68** (0.04) | **7.61** (0.06) | **9.13** (0.09) |
|     | ConvCNP   | F           | F           | F           | F           |
|     | CNP       | F           | F           | F           | F           |
|     | RCNP (sta) | **5.51** (0.21) | **6.66** (0.11) | 7.67 (0.09) | 9.35 (0.11) |
| (c) | RCNP (tor) | **5.56** (0.10) | **6.72** (0.06) | 7.70 (0.09) | 9.31 (0.14) |
|     | ConvCNP   | **5.61** (0.01) | **6.72** (0.03) | **7.78** (0.04) | **9.60** (0.08) |
|     | CNP       | 5.55 (0.03) | 6.29 (0.03) | 6.80 (0.02) | 7.52 (0.09) |

The results in Table S14 (a) extend the results discussed in the previous section (Table S13 (a)). The results indicate that when the models are trained and tested with interpolation tasks generated based on centered image data, RCNP (sta) works better than RCNP (tor). We believe this is because the relative locations calculated based on pixel coordinates on the image canvas are more informative about the absolute location than relative locations calculated on the torus. In other words, while RCNP (sta) and RCNP (tor) do not encode the absolute target location on the image canvas, RCNP (sta) can learn to derive equivalent information based on regularities between context sets.

The image interpolation task becomes translation equivariant on the torus when the models are trained or tested using the image data with random translations. The results reported in Table S14 (b) indicate that using differences calculated on the torus allows RCNP (tor) to generalize between centered and translated images while the other models are not able to make sensible predictions in this OOID task. All models work well when trained with matched data (Table S14 (c)), meaning that training on image data with random translations allows the other models to learn the equivariance which is encoded in RCNP (tor). The best results in Table S14 (c) are observed with ConvCNP, while RCNP (tor) and RCNP (sta) are similar to CNP when $N = 2$ and better than CNP when more context data is available.

### H.3.4 Computation time

The models were trained on a single GPU, and each training run with RCNP took around 3–4 hours, each training run with CNP around 1–2 hours, and each training run with ConvCNP around 4.5–5.5 hours depending on the training data. The results presented in this supplement required 110 training runs with total runtime around 350 hours. In addition other experiments carried out with image data took around 2000 hours.

