# OpenReview forum: "Practical Equivariances via Relational Conditional Neural Processes"
_NeurIPS.cc/2023/Conference — NeurIPS 2023 poster_

### Official Review · Reviewer_bQ2F · 2023-07-02

**Soundness:** 1 poor
**Presentation:** 2 fair
**Contribution:** 1 poor
**Rating:** 2
**Confidence:** 4

**Summary:**

This paper is an extension of the conditional neural process. The primary contribution is to augment the conditional neural process with the relational structure. The developed model was examined in synthetic regression, Bayesian optimization, Lotka-Volterra simulation, and others.

**Strengths:**

1. This paper is complete in organization and easy to understand.
2. The relational inductive biases mentioned in this work are crucial in some scenarios but were already studied in previous NP related work (See weakness).

**Weaknesses:**

**1. Unclear Motivation.** The motivation for using relational inductive biases is not well clarified in the Introduction section, e.g., in which scenarios or datasets the relational information is required. Meanwhile, I am not sure of the definition of relational in this work, and it seems the concept is similar to that in Set Transformer .

**2. Lack of Novelty.** The novelty of this work seems relatively limited. There have been extensive works to incorporate equivariance into neural process models, such as work in [1-4]. As for the use of relational inductive bias, especially when the input/output are high dimensional, this has appeared in work [5].

**3. Incomplete Analysis.** In Table (1)/(2), the kl divergences are reported, but I did not find what it means in the results. Notably, most empirical analysis claims the proposed model outperforms others, but the reasons behind these observations, e.g., what kind of equivariance matters, are unclear.

**4. Missing benchmarks and baselines.** Images naturally hold translation equivariance/invariance properties and are more appropriate for evaluation. This work fails to compare with other equivariant (C)NPs [1-4].


**5.** The contribution and the organization of the paper are mixed in Line 49-Line 56.

**References:**

[1] Kawano, M., Kumagai, W., Sannai, A., Iwasawa, Y., & Matsuo, Y. (2021). Group equivariant conditional neural processes. arXiv preprint arXiv:2102.08759.

[2] Holderrieth, P., Hutchinson, M. J., & Teh, Y. W. (2021, July). Equivariant learning of stochastic fields: Gaussian processes and steerable conditional neural processes. In International Conference on Machine Learning (pp. 4297-4307). PMLR.

[3] Markou, S., Requeima, J., Bruinsma, W., Vaughan, A., & Turner, R. E. (2021, October). Practical Conditional Neural Process Via Tractable Dependent Predictions. In International Conference on Learning Representations.

[4] Foong, A., Bruinsma, W., Gordon, J., Dubois, Y., Requeima, J., & Turner, R. (2020). Meta-learning stationary stochastic process prediction with convolutional neural processes. Advances in Neural Information Processing Systems, 33, 8284-8295.

[5] Wang, Q., & Van Hoof, H. (2022, June). Model-based meta reinforcement learning using graph structured surrogate models and amortized policy search. In International Conference on Machine Learning (pp. 23055-23077). PMLR.

**Questions:**

No.

**Limitations:**

Limitations are highlighted in Section 7. Here it seems that the runtime complexity of predictive distributions is higher than that of Gaussian processes, which is quadratic w.r.t. the number of data points. Remember that one of the motivations in CNPs or NP-like models is the reduction of runtime complexity.
Other limitations are included in the weakness section.
The definition of equivariance is incorrect.
Some missing citations [1-5].

---

> ### Author Rebuttal · Authors · 2023-08-09
>
> Thank you for your time and effort spent reading our paper. We would like to clarify what we perceive as misunderstandings of the nature and contribution of our paper. Our core contribution is in providing a simple, effective way to build exact equivariances (with a focus on translational equivariance) directly in the CNP architecture, *in a way that easily scales to higher input dimensions*. This is something that is currently not addressed in existing work, including the references you provided. Here we would like to clarify the perceived weaknesses of our paper, views which we do not share and explain further below.
>
> ### Weaknesses
>
> > *1. Unclear Motivation. The motivation for using relational inductive biases is not well clarified in the Introduction section, e.g., in which scenarios or datasets the relational information is required.*
>
> We clarified our motivation in the Introduction: proposing a simple method to implement some exact equivariances into CNPs (particularly, translational equivariance), in a way that scales efficiently with input dimension. The relational approach is how we achieve our goal.
>
> > *Meanwhile, I am not sure of the definition of relational in this work, and it seems the concept is similar to that in Set Transformer.*
>
> We use Set Transformers to achieve invariance to permutations of the context set (see line 128). Unlike Set Transformers, we encode data points via the comparison function $g$. We call our encoding (and method) “relational” because data points are encoded based on how they compare or relate to each other.
>
> > *2. Lack of Novelty. The novelty of this work seems relatively limited. There have been extensive works to incorporate equivariance into neural process models, such as work in [1-4]. As for the use of relational inductive bias, especially when the input/output are high dimensional, this has appeared in work [5].*
>
> We firmly disagree with this claim, which misses our main contribution. All existing methods for incorporating exact equivariances in CNPs (ConvCNP and variants [3,4], SteerCNP [2], EquivCNP [1]) require at some point a regular discretization (lattice) of the input space and convolution operations. This limits in practice the number of input dimensions these methods are applied to (often up to 1 or 2 equivariant dimensions), as lattice-based methods scale exponentially with the input dimension.
>
> Thank you for the references. Note we already had [3,4] in our paper. **We will include a more extensive discussion of [1-5] in the Related Works.** Briefly:
>
> - [2] introduces SteerCNP, which directly generalizes ConvCNP to other equivariances and suffers from its exact same scaling issues (e.g., Section 8 of [2], “*One practical limitation of this method is the necessity to discretise the continuous RKHS Embedding, which can be costly*”). [2] mentions the possibility of using alternative architectures to bypass this issue, but this is not developed in the paper.
> - [1] introduces EquivCNP, based on LieConv (Finzi et al., *ICML* 2020). While LieConv per se can operate on irregular point clouds, EquivCNPs require constructing a lattice over the input space (see Algorithm 1 and Section 4.3 of [1]), thus reinstating the scalability problem.
> - [5] aims to learn arbitrary relational inductive biases in a NP model using a relational graph structure on the context set. While interesting, this differs from our goal of directly building *exact equivariances* in the CNP architecture.
>
> > *3. Incomplete Analysis. In Table (1)/(2), the kl divergences are reported, but I did not find what it means in the results. Notably, most empirical analysis claims the proposed model outperforms others, but the reasons [...] are unclear.*
>
> Our RCNP models implement specific equivariances (translational or isotropy), as specified in each experiment. If our method works better, it is because encoding that equivariance is useful for the task; we will add an explanation.
>
> > *4. Missing benchmarks and baselines. Images naturally hold translation equivariance/invariance properties and are more appropriate for evaluation.*
>
> We added new results on MNIST and CelebA (see Table 3 and Fig 1 in the rebuttal PDF). In our paper, we focused on higher-dimensional tasks to showcase the capabilities of our model.
>
> > *This work fails to compare with other equivariant (C)NPs [1-4].*
>
> This is not correct. We compared against ConvCNP and ConvGNP [3], where applicable. Due to the mentioned scalability issues, we could not test these methods above input $d_x=2$. ConvNPs [4] have the same scalability issues as ConvCNP and are generally outperformed by ConvGNP (see [3]). The other equivariant CNPs [1,2] have the same scalability issues. Moreover, SteerCNP [2] is a strict generalization of ConvCNP, so there is no need to run separate tests in the translational equivariant case.
>
> > 5. The contribution and the organization of the paper are mixed.
>
> We rewrote a separate paragraph, distinct from the paper organization, with the contributions described at the beginning of this response.
>
> ### Limitations
>
> > *Here it seems that the runtime complexity of predictive distributions is higher than that of Gaussian processes [...].*
>
> This is not correct. Unlike CNPs which are amortized, GPs still require training (fitting the kernel parameters), which is generally *cubic* in the number of data points. There are quadratic techniques but they require tuning for good results (Maddox et al., 2021). Regarding scalability of FullRCNP, see also our answer to reviewer N64F.
>
> > *The definition of equivariance is incorrect.*
>
> Our definition of equivariance is correct and equivalent to the standard definition for general mappings, $\tau f(X) = f(\tau X)$. We added a section in the Supplement showing that our definition of equivariance of a prediction map based on the invariance of its representation (Eq. 3) is equivalent to the standard definition, as found e.g. in [2] or in the ConvCNP paper (Gordon et al., 2020).

---

> > ### Comment · Reviewer_bQ2F · 2023-08-11
> > **Major Concerns Still Exist and Keep Scores**
> >
> > Thank the author for the detailed feedback. After reading the review, I tend to keep my score since major concerns still exist, and I will explain them as follows.
> >
> > ***1. Set Transformer & Contributions & Novelty (Major Concern)***
> >
> > Note that
> >
> > (1) the author claimed "**the Set Transformer was used to achieve invariance together with a comparison function g, however, this work didn’t cite the Set Transformer at all**". As this is a core module in learning invariance and equivariance, this cannot be ignored and the contribution should be clarified in practice.
> >
> > (2) the Set Transformer can learn the equivariant map, while **the model structure of this work resembles too much on that**.
> >
> > (3) The exact equivariance is a bit vague in descriptions. Meanwhile I disagree that previous equivariant CNP work cannot scale with input dimension since the embedding function can be applied to high dimensional input.
> >
> > (4) Since the introduced modules are marginal w.r.t. the Set Transformer, I still doubt the novelty.
> >
> > ***2. Incomplete Analysis***
> >
> > In the rebuttal, the author claimed
> > >if our method works better, it is because encoding that equivariance is useful.
> >
> > (1) I didn’t see the logics of this discussion. There exist many factors to explain the increased performance, including neural architectures (particularly set transformer), model complexity and etc. (2) the meaning of kl divergence in table (1)/(2) is still not well discussed.
> >
> > ***3. High Runtime Complexity in Testing (Major Concern)***
> >
> > (1) In Appendix Line 225-227, the proposed method runtime complexity is $\mathcal{O}(MN^2)$, while the runtime complexity for CNPs is $\mathcal{O}(N+M)$ in prediction. **Note that one motivation for developing CNPs is to reduce the inference complexity of Gaussian processes. **
> >
> > (2) Considering the same runtime complexity, we cannot directly apply the Gaussian process with equivariant kernels to solve the problem, which should be more accurate in capturing equivariance. So what is the benefit of high runtime complexity in this work?
> >
> > ***4. Still Wrong Definition of Equivariance in Line72 Eq. (3)***
> >
> > Note that the output in Eq. (3) is after the $r$’s transformation; however, there is no equivariant operation w.r.t. the $r$’s output. This violates the definition of equivariance. Also, this is not the standard definition of the ConvCNP paper.

---

> > > ### Author Response · Authors · 2023-08-11
> > > **Our Further Clarifications to Major Misunderstandings**
> > >
> > > Thank you for your detailed response, to which we respond below.
> > >
> > > **1a. Set Transformer (Major Misunderstanding)**
> > >
> > > We apologize for a truly unfortunate typo in our rebuttal (not our paper), which likely emerged while trimming our response. We meant to say "We use **Deep Sets** to achieve invariance to permutations of the context set (see line 128)." The citation in line 128 is the DeepSets paper. **We do not use Set Transformer.** This can be verified by checking our code submitted in the Supplement.
> > >
> > > Set Transformers differ from DeepSets in that they add an attention mechanism which *learns* interactions between elements of the set. **Our method does not use attention.** Instead, we encode an *exact equivariance* via the comparison function $g$. Notably, this is an orthogonal point. $g$ can likely be applied with Set Transformers (instead of DeepSets) to augment our approach with attention, while at the same time encoding exact equivariances from the start. However, departing from the DeepSet architecture (as mentioned also by you later) would confuse our contribution and muddle the comparison with prior CNP work on exact equivariances, based on (Conv)DeepSets. Thank you for raising this point, which we will add to Related Works and Discussion, with references to the Set Transformer (Lee et al., 2019) and Transformer Neural Processes (Nguyen & Grover, 2022). Apologies for not having included these references before – while they do not deal with exact equivariances, they should definitely be discussed.
> > >
> > > **1b. Learning Equivariances vs. Implementing Exact Equivariances (Major Misunderstanding)**
> > >
> > > Overall, the crux seems to be a difference in views between "learning (approximate) equivariances" and "building exact equivariances in the network architecture". Our method does the latter.
> > >
> > > We stand by our statement that no existing method is able to incorporate *exact* equivariances in the CNP architecture from the start (e.g., like CNNs incorporate exact translational equivariance), in a way which scales well with input dimension. We believe that there is a fundamental conceptual difference between learning approximate equivariances and incorporating specific exact biases directly in the network architecture. Incidentally, the two approaches are not opposed; we could likely incorporate some exact equivariances and let the network learn others.
> > >
> > > **2. Analysis**
> > >
> > > **We do not use Set Transformers.** Our architecture is based on DeepSets and a fixed comparison function $g$. We aimed to keep a similar complexity between different networks in terms of e.g. number of parameters. We are happy to answer more specific questions about the analysis.
> > >
> > > The KL is computed, when possible, between the predictive map of the CNP $p(Y^*| X^*; X, Y)$ and the ground-truth posterior predictive map from the GP. We will include a clear definition, apologies that it was not explained before. This metric was used by e.g. Bruinsma et al. (2023).
> > >
> > > **3. Runtime Complexity in Testing**
> > >
> > > First, please note that the complexity for the diagonal RCNP (which, as we proved both in theory and practice, is *exact* for translational equivariance) is only $O(MN)$. This is worth considering given that translational equivariance is such an important property and plenty of CNP work on exact equivariances focuses *exclusively* on that (e.g., ConvCNP, Gordon et al. 2020; ConvGNP, Markou et al., 2022; FullConvGNP, Bruinsma et al., 2020). In practice, diagonal RCNPs can be *faster* at runtime than ConvCNP, their direct counterpart (see Table S1).
> > >
> > > Second, CNPs are amortized while GPs are not. So even if the asymptotic complexity of FullRCNP is $O(N^2M)$, due to lack of amortization, GPs will need to be trained, which is $O(N^3)$. In many applications, the model receives data sequentially (e.g., BayesOpt), which means that the cost of GP training is applied repeatedly.
> > >
> > > **4. Definition of Equivariance is Correct**
> > >
> > > Please note that we are defining equivariance of the *prediction map*. Since a prediction map fully depends on its representation $r$, *we can define equivariance of the prediction map based on properties of $r$* (i.e., invariance), there is no contradiction. We will clarify this in the paper.
> > >
> > > Our condensed proof is:
> > >
> > > The standard definition of equivariance for a generic mapping is
> > > $$\tau F_Z = F_{\tau Z}$$
> > >
> > > Starting from the definitions of the ConvCNP paper (Property 2, Gordon et al., 2019):
> > > $$\tau(X,Y) = (\tau X, Y), \qquad \tau f(X) = f(\tau^{-1} X)$$
> > >
> > > We apply the defs. above to a prediction map, seen as the function that takes as input the context set $(X, Y)$ and outputs a function that takes as input the target set $X^*$:
> > > $$P_{\tau(X,Y)} = \tau P_{(X,Y)}  \Leftrightarrow  \forall X^*, \ p(\cdot \mid r((\tau X,Y), X^*))= p(\cdot \mid r((X,Y),\tau^{-1}X^*))$$
> > >
> > > Applying $\tau^{-1}$ to each side:
> > > $$P_{\tau(X,Y)} = \tau P_{(X,Y)} \Leftrightarrow  \forall X^*, r((X,Y),X^*) = r((\tau X,Y),\tau X^*)$$
> > >
> > > which is our definition (Eq. 3).

---

### Official Review · Reviewer_jSJf · 2023-07-07

**Soundness:** 4 excellent
**Presentation:** 4 excellent
**Contribution:** 3 good
**Rating:** 7
**Confidence:** 5

**Summary:**

This work introduces a new member of the neural process model family that is designed for biasing the model towards representing equivariances in the data. It does this by including relational information among the context set, and between the predicted and context inputs in the encoder for a new input. Also, only the relational information is used for encoding and the absolute information is discarded. Experiments are given for a several types of equivariances/applications to show the method successfully models equivariances in the data.

**Strengths:**

Simple but effective technical idea with convincing experiments. For instance, I found Figure 1 very illuminating. The work shows promise for scaling GP-like models (or rather, emulating GP kernels) for distributions over functions to large data/dimensions.

**Weaknesses:**

The full RCNP variant has tractability issues, although it is shown that the simpler RCNP using the diagonal elements of the relational matrix performs satisfactorily.

Equivariances other than translations and rigid transformations, and other variants of CNP are left to future work. Would it not be too much effort to examine a few of them in this paper?


**Questions:**

Rather than inputting the full unweighted relational matrix to the embedding, would it be possible to learn which pairs of context set input/outputs and the value of their comparison function to attend to? In addition to weighting pairs of data points, some variant of attention could help break the quadratic scaling.

What is the cause of the jagged mean line for RCNP in Figure 1 (b), (e) relative to CNP and GPs?

**Limitations:**

An honest assessment of the limitations is given in Section 7, including the point that the full RCNP model has quadratic scaling.

---

> ### Author Rebuttal · Authors · 2023-08-09
>
> Thank you for your very positive comments and useful remarks about our work. We address your remarks and questions below.
>
> ### Weaknesses
>
> > *The full RCNP variant has tractability issues, although it is shown that the simpler RCNP using the diagonal elements of the relational matrix performs satisfactorily.*
>
> This is a fair point. As counterpoints to consider, we would like to remark that CNPs are commonly applied in the low-data setting (e.g., up to several hundred data points). Thus, while the quadratic scaling of full RCNPs is admittedly improvable, it is not as limiting as it might seem from its asymptotic analysis.
> Moreover, as you mentioned, for the very common application of translational invariance, “diagonal” RCNPs are enough. Importantly, for the case of translational equivariance, the diagonal RCNP is not just an approximation of the full RCNP, but an exact solution. This is proved in our theorems (see Proposition 4.9 in the main text and its proof in Appendix B.2), and confirmed by our empirical results.
>
> > *Equivariances other than translations and rigid transformations, and other variants of CNP are left to future work. Would it not be too much effort to examine a few of them in this paper?*
>
> Good point. In terms of examining other variants of (R)CNP, we implemented and ran new experiments with the autoregressive RCNP (AR-RCNP), where the AR-CNP is a recent model introduced in Bruinsma et al. (2023). The AR-RCNP demonstrates the application of our technique to another model of the CNP family. Results are presented in Table 1 of the rebuttal PDF, and will be included in the revised paper.
>
> Regarding other equivariances: we now also implemented equivariance to proper and improper rotations. Please refer to our answer to reviewer C6E4 for a detailed description of the newly implemented model, experiments and related discussion. We report a summary below.
>
> The newly implemented equivariance is based on the comparison function:
> $$g_\text{rot}(\mathbf{x},\mathbf{x}') = (\Vert \mathbf{x} - \mathbf{x}' \Vert_2, \Vert \\mathbf{x} \Vert_2, \Vert \mathbf{x}' \Vert_2),$$
> which is based on the distance between points as well as their distance from the origin (the center of rotation). This makes the comparison function invariant to rotations and mirroring, but not translations. Thus, it will induce a FullRCNP which is equivariant to these transformations, according to our Proposition 4.5.
>
> We tested this new RCNP variant in a new set of synthetic experiments which incorporate rotational and mirror symmetry. Preliminary results are presented in Table 2 of the rebuttal PDF, and full results will be included in the revised paper and Supplementary Material. Our results show that the FullRCNP is able to leverage the equivariances intrinsic to the task to outperform a standard CNP in both low and high dimensions.
>
> In conclusion, our framework is suitable for isotropy, translational equivariance, and the newly implemented (proper and improper) rotational equivariance. Whether other equivariances can be expressed via our relational approach is an interesting direction for future work. Nonetheless, we believe the equivariances addressed in the paper represent a large class of useful equivariances. This is demonstrated by the fact that several key papers and models proposed in the neural process literature (as well as the broader machine learning literature) focus *only* on translational equivariance (e.g., ConvCNP, Gordon et al. 2020; ConvGNP, Markou et al., 2022; FullConvGNP, Bruinsma et al., 2020), with the limitations we discussed in the paper and that our proposed method overcomes.
>
>
> ### Questions
>
> > *Rather than inputting the full unweighted relational matrix to the embedding, would it be possible to learn which pairs of context set input/outputs and the value of their comparison function to attend to? In addition to weighting pairs of data points, some variant of attention could help break the quadratic scaling.*
>
> This is an interesting point and related to work on Transformer neural processes (Nguyen & Grover, 2022). However, we are unsure how this would address the scaling issue per se, since the attention mechanism is also notoriously quadratic in the size of the context set.
>
> A somewhat similar idea, along these lines of finding a middle ground between the full RCNP and the diagonal RCNP, would consist of something like a low-rank approximation of the full comparison set, or by only comparing to a chosen subset of K "important" context points. This would relate to work in the Gaussian process literature (“sparse” GPs and inducing points), and also to the concept of coresets. Indeed, this is a potential direction of future work that we have been considering. Please see our response to reviewer N64F for additional remarks on this point.
>
> > *What is the cause of the jagged mean line for RCNP in Figure 1 (b), (e) relative to CNP and GPs?*
>
> - The jagged mean is a byproduct of the common ReLU activation function used in the (R)CNP architecture. Due to the ReLUs, the output of the network will naturally be a piecewise linear function (i.e., with discontinuous derivatives, which makes it look jagged).
> - Standard CNPs here underfit the data, so their mean *appears* smoother. However, note that it is still jagged after zooming in – the discontinuous derivative is just less prominent due to the underfitting. The outputs of CNPs commonly look jagged (e.g., see Figure 2 of Garnelo et al. 2018).
> - GPs are kernel methods, here with a Matérn 5/2 kernel which is twice differentiable, so the posterior mean function will also be twice differentiable (i.e., it will look reasonably smooth).
>
> ### References
>
> - Bruinsma et al. (2023). Autoregressive Conditional Neural Processes. *ICLR*.
> - Garnelo et al. (2018). Conditional Neural Processes. *ICML*.
> - Nguyen & Grover (2022). Transformer neural processes: Uncertainty-aware meta learning via sequence modeling. *ICML*.

---

### Official Review · Reviewer_N64F · 2023-07-09

**Soundness:** 4 excellent
**Presentation:** 3 good
**Contribution:** 4 excellent
**Rating:** 8
**Confidence:** 4

**Summary:**

The paper presents a novel approach for incorporating equivariance into conditional neural processes (CNPs) which can scale to high dimensions. Modelling equivariance is essential to improve the performance of CNPs. Unlike previous approaches that use convolution and become impractical with increased input dimensions, this work uses relational information and discards absolute information. As a result, this simple method can handle high-dimensional inputs. The authors also prove that their approach is context preserving which means they do not lose other information. Their empirical results demonstrate that the proposed method is comparable to convolutional CNPs, GNPs etc. on a diversified range of tasks.


**Strengths:**

**Strengths**

1. This paper addresses the challenge of incorporating equivariance into Conditional Neural Processes (CNPs) for high-dimensional problems. The proposed models are shown to be translation-equivariant, allowing them to scale to higher dimensions and are comparable/outperform existing CNP and GNP models on a wide range of tasks.
2. The paper is well-structured and clearly presented, with a strong motivation for the research, precise technical statements, and comprehensive background.
3. The paper provides robust theoretical results, all of which are supported by proof. The empirical investigation is extensive, covering a diversified range of tasks. The authors demonstrate the effectiveness of their models through experiments on synthetic Gaussian and non-Gaussian regression tasks, Bayesian optimization, Lotka-Volterra models and reaction-diffusion models.

**Weaknesses:**

**Weakness**

Given the extensive use of RGNP in the experiments and its significant role in the paper's findings, it would still be beneficial for the authors to provide a concise description or a mathematical formulation of the RGNP in the main text. Even though it is not difficult to extend the mathematical description of RCNP to RGNP.

**Questions:**

**Questions**

1. I wonder if there is a middle ground between the full RCNP and the diagonal RCNP. In the matrix terminology, it is a bit similar to a low-rank approximation of the full comparison set. I think this would allow people to balance computational cost and expressivity. Also by only comparing only to a subset of K "important" context points, we can lower the cost to O(NK) rather than the quadratic cost of the full RCNP.
2. By only using pairwise comparison in the model construction, I wonder whether we would lose high-order interaction which involves more than two context points.

**Limitations:**

The authors have discussed the limitations of their work and I have no concerns of any potential negative societal impact.

---

> ### Author Rebuttal · Authors · 2023-08-09
>
> Thank you for your very positive and insightful comments, we are glad to see that you find our paper particularly well-suited for NeurIPS. In the following we address your questions and points raised.
>
> ### Weaknesses
>
> > *Given the extensive use of RGNP in the experiments and its significant role in the paper's findings, it would still be beneficial for the authors to provide a concise description or a mathematical formulation of the RGNP in the main text. Even though it is not difficult to extend the mathematical description of RCNP to RGNP.*
>
> Thanks for pointing this out. In the revised paper we will include a concise mathematical description of the RGNP model in the main text.
>
> ### Questions
>
> > *1. I wonder if there is a middle ground between the full RCNP and the diagonal RCNP [...] similar to a low-rank approximation of the full comparison set. [...] Also by only comparing only to a subset of K "important" context points [...].*
>
> This is a very interesting and insightful suggestion to lower the asymptotic cost of full RCNPs, which would be related to other work in the Gaussian process literature (“sparse” GPs and inducing points), and also to the concept of coresets. Indeed, this is a potential direction of future work aimed at reducing the computational complexity.
>
> A naive baseline to consider in this direction would be to only encode a target point based on the $K$ closest context points, thus reducing the cost of FullRCNP to $O(K^2 M + N M)$, where the first term is the relational encoding and the second term is the naive cost of computing the distances between $M$ targets and $N$ context points (although the latter can be sped up via smarter nearest-neighbor search algorithms; see e.g. Hyvönen et al. 2022). While $K$ is nominally a constant, to guarantee a reasonable performance in practice we envision it could be chosen based on the (maximum) number of context points in the task. Still, by choosing a square-root scaling ($K \propto \sqrt{N}$) we can recover a manageable asymptotic cost of $O(N M)$.
>
> > *2. By only using pairwise comparison in the model construction, I wonder whether we would lose high-order interaction which involves more than two context points.*
>
> This is an excellent point and the perhaps surprising theoretical answer is no, in our construction. Our context-preservation theorems demonstrate that no information is lost about the entire context set, so in principle a RCNP with a sufficiently large network is able to reconstruct any high-order interaction of the context set (see Section 4.2 in the main text and the full proof in Appendix B.2).
>
> However, in practice the answer can be somewhere in the middle, in that the chosen representation is built on two-point interactions, so depending on the network size it may be harder for the network to effectively encode the simultaneous interaction of many context points.
>
> We will add these informative remarks in the theoretical section of the Supplement.
>
> ### References
>
> - Hyvönen et al. (2022). A Multilabel Classification Framework for Approximate Nearest Neighbor Search. *NeurIPS*.

---

> > ### Comment · Reviewer_N64F · 2023-08-10
> >
> > Thanks for the detailed response. My questions have been addressed and I strongly support the acceptance of this paper.

---

### Official Review · Reviewer_C6E4 · 2023-07-11

**Soundness:** 3 good
**Presentation:** 3 good
**Contribution:** 3 good
**Rating:** 6
**Confidence:** 3

**Summary:**

The authors propose a class of neural processes that can be constructed to enforce invariance to particular properties like translation and rotation.

**Strengths:**

- As far as I know, the proposed architecture and technique for enforcing invariances in Section 3 is novel.
- Improvements are shown over standard neural processes on a range of tasks
- The authors show both theoretically and empirically that the proposed architecture does indeed enforce the invariances described.

**Weaknesses:**

- Choosing $g$ in Eq 5 to enforce a particular invariance seems difficult. The authors provide choices that enforce isotropy and translation invariance, but it would not be obvious how to enforce a different type of invariance.

**Questions:**

See weaknesses

**Limitations:**

Limitations are adequately addressed

---

> ### Author Rebuttal · Authors · 2023-08-09
>
> Thank you for your positive review and comments, and we are glad to see that you found our paper interesting.
>
> Regarding your concern about enforcing particular invariances, we would like to comment that it is indeed true that our method only applies to equivariances that can be enforced via a comparison function $g$. This is explicitly mentioned within the paper, such as the Introduction:
>
> > Our proposed method works for equivariances that can be expressed relationally via comparison between pairs of points (e.g., their difference or distance); [...]
>
> and the Discussion (just before the *Limitations* paragraph):
>
> > Our method applies to equivariances that can be induced via an appropriate comparison function; here we focused on translational equivariances (induced by the difference comparison) and equivariances to rigid transformations (induced by the distance comparison).
>
> To further address your point and provide an additional example, we ran a new set of experiments with a FullRCNP which implements equivariance to proper and improper rotations (i.e., rotations + mirroring). As per Proposition 4.5 of our paper, we need to select an adequate comparison function, which we define as follows:
> $$g_\text{rot}(\mathbf{x},\mathbf{x}') = (\Vert \mathbf{x} - \mathbf{x}' \Vert_2, \Vert \\mathbf{x} \Vert_2, \Vert \mathbf{x}' \Vert_2)$$
> This comparison function is based on the distance between points as well as their distance from the origin (the center of rotation). This representation makes the comparison function invariant to rotations and mirroring, but not translations. Thus, it will induce a FullRCNP which is equivariant to these transformations.
>
> As mentioned in the global rebuttal, we tested this new RCNP variant in a new set of synthetic experiments, namely a regression task involving a Gaussian Process whose mean and covariance functions introduce rotational and mirror symmetry, but not translational symmetry. Such a task is representative for example of a physical model set in a potential well (e.g., a point charge or mass in the origin). Due to the limited time available for the rebuttal, we present preliminary results in Table 2 of the rebuttal PDF. Full results will be included in the revised paper and Supplementary Material. Briefly, our results show that the FullRCNP is able to leverage the equivariances intrinsic to the task to outperform a standard CNP in both low and high dimensions.
>
> In conclusion, our relational neural process framework is suitable for isotropy, translational equivariance, and the newly implemented (proper and improper) rotational equivariance. Whether other equivariances can be expressed via our relational approach is an interesting direction for future work. Nonetheless, we believe the equivariances addressed in the paper represent a large class of useful equivariances. This is demonstrated by the fact that several key papers and models proposed in the neural process literature (as well as the broader machine learning literature) focus *only* on translational equivariance (e.g., ConvCNP, Gordon et al. 2020; ConvGNP, Markou et al., 2022; FullConvGNP, Bruinsma et al., 2020), with the limitations we discussed in the paper and that our proposed method overcomes.
>
> For clarity, we will include an additional explanation based on the paragraph above to the *Limitations* section of the paper to make the contribution and limitations of the paper clearer.
>
> ### References
>
> - Bruinsma et al. (2020). The Gaussian Neural Process. *AABI*.
> - Gordon et al. (2020). Convolutional Conditional Neural Processes. *ICLR*.
> - Markou et al. (2022). Practical Conditional Neural Processes via Tractable Dependent Predictions. *ICLR*.

---

### Author Rebuttal · Authors · 2023-08-09

We thank the anonymous reviewers for their comments and suggestions for improving our paper. We are glad to see that the majority of reviewers found the paper interesting and of impact. We provide clarifications and detailed answers to perceived weaknesses and raised questions in our individual responses.

In particular, we added a number of requested experiments and comparisons to baselines in different settings to the paper. Tables and Figures presenting these results can be found in the attached pdf. Please find below a short overview of the implemented changes:

### New experiments and results

- As asked by reviewer jSJf, we implemented and ran new experiments with another variant of RCNP – the autoregressive RCNP (AR-RCNP), where the AR-CNP is a recent model introduced in Bruinsma et al. (2023). The AR-RCNP demonstrates the application of our technique to another model of the CNP family. Results are presented in Table 1 of the rebuttal PDF, and will be included in the revised paper.
- As suggested by reviewers C6E4 and jSJf, we provided the RCNP implementation of a new class of equivariances, namely proper and improper rotations. We tested this new RCNP variant in a new set of synthetic experiments, a regression task involving a Gaussian Process whose mean and covariance function introduce rotational and mirror symmetry, but not translation. Such a task is representative for example of a physical model with a potential well. Due to the limited time available for the rebuttal, we present preliminary results in Table 2 of the rebuttal PDF. Full results will be included in the revised paper and Supplementary Material. Please also see our response to reviewer C6E4 for further details.
- As requested by reviewer bQ2F, we ran our RCNP model on an image completion task using classic image datasets such as MNIST and CelebA, to compare our performance to other members of the CNP family. Due to the limited time available for the rebuttal, we could only complete preliminary experiments with downscaled images. Results are presented in Table 3 of the rebuttal PDF, with examples in Figure 1 of the PDF, and will be included in the revised Supplementary Material.

### Other changes

Besides the new results explained above, we implemented a number of changes to the main text and Supplementary Material to further improve the clarity of the paper.

- We clarified our motivation and contributions (see our response to reviewer bQ2F for details).
- We expanded the Related Works section with new references and explained more in detail how our work differs from the existing literature on equivariant (C)NPs (please see our answer to reviewer bQ2F). Importantly, we highlighted how all existing works require a lattice (regular grid) construction in the input space, often followed by convolutions, which strongly limits scalability of the methods to higher input dimensions; a limitation which is not shared by our distinct “relational” approach.
- We expanded our limitations section, to better highlight that our relational approach hinges on finding an appropriate comparison function $g$. While our paper covers several important equivariances of large practical interest (translational, isotropy, now rotations), whether other equivariances can be expressed via our relational approach is left for future work. For details, see the response to reviewer C6E4.
- In addressing a remark by reviewer bQ2F, we clarified our definition of equivariance of prediction maps, based on the invariance of their representation (Eq. 3 in the paper). Our definition, used for its convenience in our paper, is mathematically equivalent to the more common definition of equivariance for general mappings; $\tau f(X) = f(\tau X)$, but the comment made us realize it is not immediately apparent. We added a section in the Supplementary Material explicitly showing this equivalence, starting from standard definitions of equivariance and group action (Gordon et al., 2020; Holderrieth et al., 2021).

### References

- Bruinsma et al. (2023). Autoregressive Conditional Neural Processes. *ICLR*.
- Gordon et al. (2020). Convolutional Conditional Neural Processes. *ICLR*.
- Holderrieth et al. (2021) Equivariant Learning of Stochastic Fields: Gaussian Processes and Steerable Conditional Neural Processes. *ICML*.

---

### Decision · Program_Chairs · 2023-09-21

**Decision:**

Accept (poster)

**Comment:**

This work proposes the relational conditional neural process, a new neural process that biases the model towards representing exact equivariances in the data. Unlike previous approaches that use convolution, this approach uses relational information instead of absolute information and so scales well to high-dimensional inputs.

Strengths
- Good presentation and writing.
- Simple and effective idea.
- Convincing experiments compared to existing CNP/GNP models and robust theoretical results.

Weaknesses
- It's unclear how to enforce other types of invariance with a comparison function g.
- The full RCNP has tractability issues though the simpler one still performs well.
- There's no concise description of RGNP in the main text.